# An experimental study of drainage network development by surface and subsurface flow in low-gradient landscapes

Brian G. Sockness[1], Karen B. Gran[1]

[1]Department of Earth and Environmental Sciences, University of Minnesota Duluth, Duluth, MN, USA

*Correspondence to*: Karen B. Gran (kgran@d.umn.edu)

**Abstract.** How do channel networks develop in low-gradient, poorly-drained landscapes? Rivers form elaborate drainage networks with morphologies that express the unique environments in which they developed, yet we lack an understanding of what drives channel development in low-gradient landscapes like those left behind in the wake of continental glaciation. To better understand what controls the erosional processes allowing channel growth and integration of surface water non-
contributing areas (NCA) over time, we conducted a series of experiments in a small-scale drainage basin. By varying substrate and precipitation, we could vary the partitioning of flow between the surface and subsurface, impacting erosional processes. Two different channel head morphologies developed, interpreted as channel growth via overland flow and seepage erosion. Channel growth was dominated by overland flow vs. seepage erosion depending on substrate composition, rainfall rate, and drainage basin relief. Seepage-driven erosion was favored in substrates with higher infiltration rates, while overland flow was
more dominant in experiments with high precipitation rates, although both processes occurred in all runs. Overland flow channels formed at the onset of experiments and expanded over a majority of the basin area, forming broad dendritic networks. Large surface water contributing areas supported numerous first-order channels, allowing for more rapid integration of NCA than through seepage erosion. When overland flow was the dominant process, channels integrated NCA at a similar, consistent rate under all experimental conditions. Seepage erosion began later in experiments after channels had incised enough for
exfiltrating subsurface flow to initiate mass wasting of headwalls. Periodic mass wasting of channel heads caused them to assume an amphitheater-shaped morphology. Seepage allowed for channel heads to expand with smaller surface water contributing areas (CA) than overland flow channels, allowing for network expansion to continue even with low surface CA. Seepage-driven channel heads integrated NCA more slowly than channel heads dominated by overland flow, but average erosion rates in channels extending through seepage erosion were higher. The experimental results provide insight into
drainage networks that formed throughout areas affected by continental glaciation, and highlight the importance of subsurface hydrologic connections in integrating and expanding drainage networks over time in these low-gradient landscapes.

## 1 Introduction

Drainage networks form in settings with distinct geologic, climatic, and relief characteristics that largely control their development over long timescales (Schumm, 1981; Schumm and Lichty, 1965). Most research efforts exploring drainage

network evolution have focused on networks in high-gradient settings (Altin and Altin, 2011; Babault et al., 2012; Castelltort and Simpson, 2006; Daag, 2003; Daag and Van Westen, 1996; Garcia and Hérail, 2005; Hovius et al., 1998; Janda et al., 1984; Maroukian et al., 2008; Simon, 1999; Winterberg and Willett, 2019). Low-gradient drainage networks are likely controlled by similar factors, but fewer studies have investigated their long-term evolution. One barrier to drainage network development is that rivers have to incorporate substantial amounts of internally-drained areas without any surface water connections, referred

to as non-contributing areas (NCAs), into their watersheds to expand. The processes by which those NCAs are integrated into the drainage network may vary between high-gradient and low-gradient upland settings.

Widespread, low-gradient uplands with abundant NCAs are common in regions impacted by continental glaciation. In the Central Lowlands physiographic region of the United States, for example, multiple advances of the Laurentide Ice Sheet during the Pleistocene scoured and deposited sediment across the region, reworking pre-existing river systems by damming, re-

routing, or filling in channels. Following glaciation, new drainage networks developed in the glacial deposits. In a classic study, Ruhe (1952) observed the gradual reestablishment of drainage networks in Iowa, USA, where younger, more recently glaciated surfaces had less extensive drainage networks than surfaces associated with earlier glaciations [Fig. 1]. Clearly network development is occurring across these low-gradient uplands in Iowa and across the region over tens of thousands of years, however we lack a process-based understanding of how integration proceeds in low-gradient landscapes with abundant

NCA.

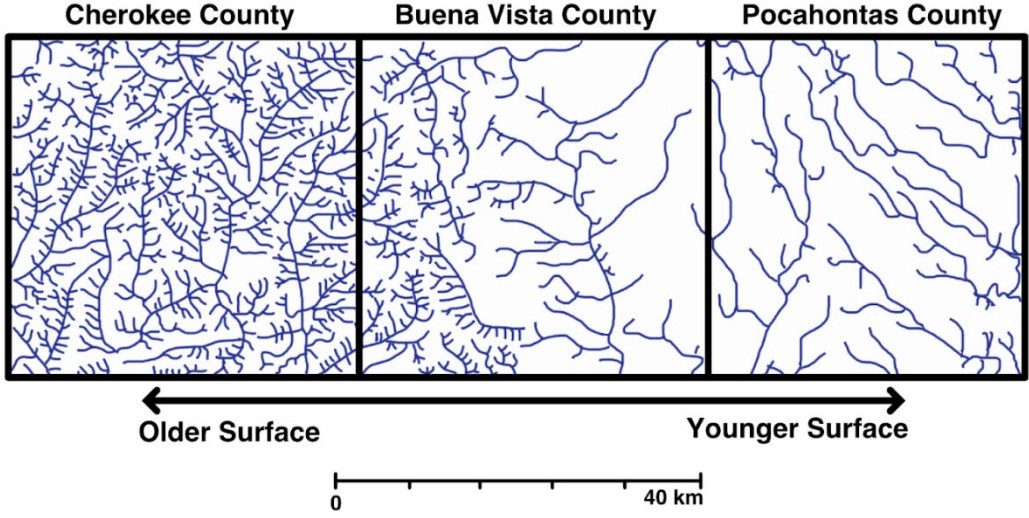

**Figure 1. Drainage network development across three counties in northwestern Iowa, USA. Glacial deposits in Cherokee County were deposited during earlier glacial periods than deposits in Buena Vista and Pocahontas counties. Higher drainage densities occur on the older deposits compared to the younger deposits (modified from Ruhe, 1952).**


There are multiple ways by which rivers can capture NCA. One entails base level fall instigating headward erosion of channels as knickpoints propagate into the uplands: a bottom-up model of drainage network development. Headward erosion incorporates NCA into the drainage network by breaching the shallow drainage divides that isolate depressions. Studies conducted in low-gradient upland settings have found that base level fall can help initiate channel incision, generate relief, and

perpetuate headward growth (Clayton and Moran, 1982; D'Alpaos et al., 2005, 2007; Fagherazzi et al., 2012; Gran et al., 2009, 2013; Matsch, 1983; Whipple et al., 2017). One of the limiting factors with bottom-up integration is that upstream area must be able to provide enough water at the channel tips to initiate erosion, a challenging condition in low-gradient terrains where substantial parts of upland surface are internally-drained.

A second method of network expansion takes more of a top-down approach, driven by connections of surface water from

NCAs associated with spillover events during periods of high precipitation, or by subsurface water from NCAs to downstream channel heads. Spillover events can be transient, leading to dynamically-variable connectivity between NCA and downstream waters (Brooks et al., 2018; Leibowitz et al., 2016; Leibowitz and Vining, 2003; Rosenberry and Winter, 1997; Shaw et al., 2012; Stichling and Blackwell, 1957), or spillover events can incise a channel to create a permanent connection between NCA and the drainage network (Douglass et al., 2009; Douglass and Schmeeckle, 2007; Hilgendorf et al., 2020). Hydrologic

connections can also occur when groundwater flows from depressions to adjacent streams, driven by the contrasting hydraulic conductivities of the region's glacial deposits (Labaugh et al., 1998; Neff and Rosenberry, 2018; Winter, 1999) or by regional groundwater flow patterns that allow subsurface flow to deviate from topographic divides and provide additional water to channels. If water contributed from surface NCA via the subsurface is able to erode channel tips through seepage erosion, then network integration can proceed via subsurface connections even in the absence of surface water connections.

The hydrologic subsidy provided by surface and subsurface connections between NCA and channels can have important implications for the long-term development of drainage networks (Lai and Anders, 2018; Hilgendorf et al., 2020; Cullen et al., 2021). If NCA are geographically isolated (Tiner, 2003) but not hydrologically isolated, then hydrologic contributions via the surface or subsurface can help integrate drainage networks. Numerical modeling by Lai and Anders (2018) showed that hydrologically-connected NCA are necessary to drive drainage network development in low-gradient landscapes. An

important, unresolved issue is how water routed via the surface or subsurface to varying degrees drive different processes of channel development and how that partitioning affects NCA integration. Cullen et al., (2021) explored the partitioning of surface and subsurface connectivity on network growth in low-gradient systems numerically and found that channel network growth was sensitive to groundwater contributions to channel heads. Geology, climate, vegetation, and relief differ throughout post-glacial landscapes of the Central Lowlands, which may favor surface or subsurface routing of potential NCA

contributions. Deconvolving the impacts of these different variables is challenging in the field, particularly given recent

anthropogenic impacts on these same post-glacial landscapes that have changed hydrologic connectivity (Foufoula-Georgiou et al., 2015; Schottler et al., 2014).

To better understand the processes that drive drainage integration via both surface and subsurface flow, we present the results from a series of drainage network evolution experiments. The experiments subjected an initially flat, internally-drained surface to rainfall and continuous base level fall to incise channels through headward erosion. We tested different combinations of substrate composition and rainfall rates to investigate how these attributes mediate the partitioning of precipitation between the surface and subsurface, driving different processes of channel development. A terrestrial lidar scanner captured high-resolution topographic data of the developing drainage network to characterize channel development, the evolution of CA and NCA through time, and the rates and patterns of network growth. Our results show that overland flow and seepage erosion drove channel development to different extents based on experimental conditions that impacted infiltration capacity, rainfall delivery rate, and relief. The experiments provide insight into the processes by which drainage networks grow and highlight the importance of subsurface flow for drainage network growth in low-gradient landscapes.

## 2 Background

### 2.1 Processes of Channel Development

Water moving through and across landscapes forms channels by exerting sufficient force to entrain and erode sediment. Overland flow exerts shear stress on the surface as a function of slope and water depth. Erosion of channel heads that occurs due to concentration of flow and steeper slopes can lead to drainage-head erosion and network expansion. In addition, shallow subsurface or groundwater flow can create or grow channels when water emerges from the subsurface with enough force to cause seepage erosion (Dunne, 1990). Erosion via seepage is a function of hydraulic gradient and permeability of substrate. Larger hydraulic gradients increase seepage forces, which can occur if groundwater recharge is greater or the interface between the surface and subsurface has greater relief (Dunne, 1980, 1990). As channels expand by seepage erosion, groundwater flow further concentrates at the channel heads and begets more erosion by positive feedback (Dunne, 1990; Cullen et al., 2021). Erosion at channel heads introduces asymmetries in the concentrated flow of groundwater, causing the direction of channel growth to adjust towards maintaining symmetrical flow (Cohen et al., 2015). The gradual erosion of sediment by seepage can eventually cause mass wasting by undermining the overlying material and eroding large volumes of sediment.

Seepage erosion has been studied at different spatial scales as a form of channel development. At large scales, seepage erosion has been attributed as the primary driver of channel development for drainage networks in unconsolidated materials (Coelho Netto et al., 1988; Micallef et al., 2021; Pillans, 1985; Schumm and Phillips, 1986; Uchupi and Oldale, 1994), and in bedrock in places like the Colorado Plateau (Howard, 1988; Laity and Malin, 1985) and Florida Panhandle (Schumm et al., 1995). The channel heads of these networks are often described as "amphitheater-shaped" due to the distinctive high relief headwalls that

form when seepage erosion undermines channel headwalls and causes mass wasting (Laity and Malin, 1985), although this morphology may arise from any curvature-driven mechanical process (Petroff et al., 2018). Seepage erosion has also been linked to distinct longitudinal profiles (Devauchelle et al., 2011) and bifurcation angles (Petroff et al., 2013; Devauchelle et al., 2012), the latter being more prevalent in regions with humid climates favoring groundwater flow to streams (Seybold et al., 2017, 2018). At smaller scales, seepage can drive gully erosion in relatively low-gradient agricultural settings (Castillo and Gómez, 2016).

The partitioning of flow to the surface vs. subsurface is largely a balance between water delivery to the surface by precipitation and water losses by infiltration into the subsurface. Determining this balance is complex because many factors controlling infiltration and evapotranspiration like vegetation type and density, substrate texture and saturation level, and topographic roughness are to some extent codependent on the precipitation rates and volumes set by the prevailing climate. Numerical models, physical experiments, and field-based studies are particularly useful approaches for determining the interactions and feedbacks between different subsets of these factors and their influence on infiltration and runoff generation (Berhanu et al., 2012; Dunne et al., 1991; Huang et al., 2013; Lobkovsky et al., 2004; Morbidelli et al., 2015; Mu et al., 2015; Nassif and Wilson, 1975; Schorghofer et al., 2004; Thompson et al., 2010).

For this study, the effects of sediment texture and precipitation rates on infiltration and flow pathways in low-gradient upland settings are studied. In isolation, coarse-grained sediments have greater infiltration capacities than fine-grained sediments, allowing precipitation to infiltrate faster, potentially reducing the degree of surface water ponding. Also in isolation, greater rainfall rates provide larger volumes of water over a given timespan, increasing the likelihood of attaining saturation, surface water ponding, and overland flow. However, the combined effects of slope, substrate texture, and rainfall rates on flow pathways remain difficult to determine, as reviewed by Morbidelli et al. (2015). They suggest that interactions between surface and subsurface water may be an important and largely unresolved factor controlling infiltration across different slopes, making it important to consider processes associated with both surface and subsurface water.

## 2.2 Previous Drainage Network Development Experiments

Physical experiments conducted in the laboratory allow us to study channel development under controlled conditions and reduced spatial scales. The apparatuses used to model channel development have typically incorporated three fundamental design elements: an erodible substrate, a precipitation source, and a mechanism to adjust base level. These elements simulate three of the major controls of drainage network development: geology, climate, and tectonics, respectively. Prior experiments have investigated how changing these conditions affects the processes of drainage network development on an initially unchannelized surface (Berhanu et al., 2012; Bonnet and Crave, 2003; Hasbargen and Paola, 2000; Lague et al., 2003;

Lobkovsky et al., 2004; Parker, 1977; Pelletier, 2003; Phillips and Schumm, 1987; Schorghofer et al., 2004; Singh et al., 2015; Smith et al., 2008; Sweeney et al., 2015).

Parker (1977) showed that channel network development by overland flow followed the temporal phases of initiation, elongation, and elaboration first proposed by Glock (1931). Pelletier (2003) built on these results by testing channel network growth under different topographic configurations. Similar to other studies (Phillips and Schumm, 1987), they found that overland flow produced dendritic drainage networks at a rate dependent on the initial slope of a planar surface. However, convex plateau-like surfaces had a combination of channelization by both overland flow and seepage erosion.

Other experiments have shown how the development of drainage networks by overland flow can result in different steady-state topography under constant uplift and precipitation (Bonnet and Crave, 2003; Hasbargen and Paola, 2000; Lague et al., 2003). Lague et al. (2003) found that internally-drained areas were captured at an exponential rate before fully integrating the initial surface. Increasing the uplift rate caused the mean elevation to increase throughout the basin (Bonnet and Crave, 2003; Lague et al., 2003) and channel morphology adjusted to have a smaller cross-sectional area (Turowski et al., 2006). Ouchi (2011) described an episodic "erosion with knickpoints" mode of fluvial erosion that steepened slopes on uplifted surfaces versus a continuous "erosion of declining slope" that decreased slopes when relief was low. Recent efforts have emphasized the role of hillslope processes that act with channel-forming processes in creating steady-state landscape morphologies (Singh et al., 2015; Sweeney et al., 2015).

Experiments have also focused on channel network growth by seepage erosion. Howard and McLane (1988) allowed groundwater from an adjacent reservoir to move through a package of sediment and exfiltrate through a sloping valley wall. They observed that seepage erosion was strongest at a narrow band where groundwater exfiltrated from the valley wall and undermined the overlying sediment. The overall rate of channel growth by seepage erosion was limited in these experiments by the ability of fluvial transport to remove material from the valley floor after a mass wasting event. Howard (1988) performed similar experiments with slightly cohesive sediment and found that seepage erosion produced narrower and more incised channels. Lobkovsky (2004) showed that seepage erosion is slope-dependent and that beyond a critical slope angle, it can mobilize sediment at slopes less than its maximum angle of stability. Gomez and Mullen (1992) augmented these approaches by using precipitation rather than an adjacent reservoir. They found that headward growth of drainage networks by seepage erosion proceeded in phases similar to what Parker (1977) described for overland flow, but with a different channel morphology. Berhanu (2012) showed that seepage erosion driven by rainfall produced wider, bifurcated channels compared to single, elongated channels produced by groundwater flowing unidirectionally from an adjacent reservoir. The experiments discussed here augment these earlier efforts by investigating the conditions necessary for erosion via surface vs. subsurface flow, with a specific focus on the interplay of overland flow vs. seepage erosion on rates of erosion, integration of NCA, and network expansion in low-gradient landscapes.

## 3. Methodology

### 3.1 Drainage Network Evolution Experiments

We performed a series of small-scale experiments to simulate the development of drainage networks. The focus of the experiments was to evaluate how precipitation rates and substrate compositions mediate the processes and rates of drainage network development. To do this, we conducted six experiments where channel development was observed from genesis to full elaboration (10-14 hours) under a range of rainfall rates and substrate compositions [Table 1; Table 2].

Topographic data were captured at discrete time intervals using a FARO Focus 3D terrestrial laser scanner suspended 1 m above the basin. The position of the scanner relative to the basin surface provided point cloud data with a point spacing of 2 mm. The scanner was positioned in the same location for each scan using a computer-controlled cart set on tracks above the basin. Both rainfall and base level fall ceased for approximately ten minutes while positioning and capturing each scan.

**Table 1. Experimental conditions, duration, and scan intervals used for each experimental run.**

| Run | Substrate Clay Fraction | Rainfall Rate (R) | Uplift Rate (U)[†] | Infiltration Capacity (I) | U/R Ratio | I/R Ratio | Run Duration | Scan Interval |
|-----|-----|-----|-----|-----|-----|-----|-----|-----|
| - | weight % | µm/s | µm/s | µm/s | - | - | hr | hr |
| 1 | 0 | 11 | 3.2 | 310 | 0.3 | 28 | 13 | 3,2[*] |
| 2 | 2 | 16 | 3.2 | 132 | 0.3 | 8 | 10 | 2 |
| 3 | 2 | 8 | 3.2 | 132 | 0.4 | 17 | 14 | 2 |
| 4 | 2 | 16 | 3.2 | 132 | 0.2 | 8 | 14 | 2 |
| 5 | 6 | 16 | 3.2 | 26 | 0.2 | 2 | 14 | 2 |
| 6 | 6 | 8 | 3.2 | 26 | 0.4 | 3 | 14 | 2 |

[*]Scan interval of three hours for the first scan and two hours for all subsequent scans

[†]Uplift rate is equivalent to base level fall rate.

**Table 2. Labels used to refer to the substrate clay fractions and rainfall rates used in experiments.**

| Substrate Clay Fraction | Label | Rainfall Rate | Label |
|-----|-----|-----|-----|
| weight % | - | µm/s | - |
| 0 | No Clay | 8 | Low Rainfall |
| 2 | Moderate Clay | 11 | Moderate Rainfall |
| 6 | High Clay | 16 | High Rainfall |

Experiments were conducted in a 0.95 m tall by 0.80 m diameter cylindrical drum designed by Gazzetti (2015) after the apparatus used by Hasbargen and Paola (2000) [Fig. 2]. The drum holds sediment exposed to rainfall under a constant rate of base level fall to generate a drainage network. Base level fall at the outlet has the same effect as uplift of the basin. A 0.02 m wide outlet spans the height of the drum where sediment and water discharge from the basin [Fig. 2b]. A computer-controlled step motor lowers a metal gate at the outlet, dropping base level and instigating channel incision into the substrate. These

experiments used a constant base level fall (e.g. uplift) rate of 1.15 cm/h that was equivalent to Gazzetti's (2015) "low" uplift rate and slightly faster than Hasbargen and Paola's (2000) rate of 1.00 cm/h [Fig. 2].

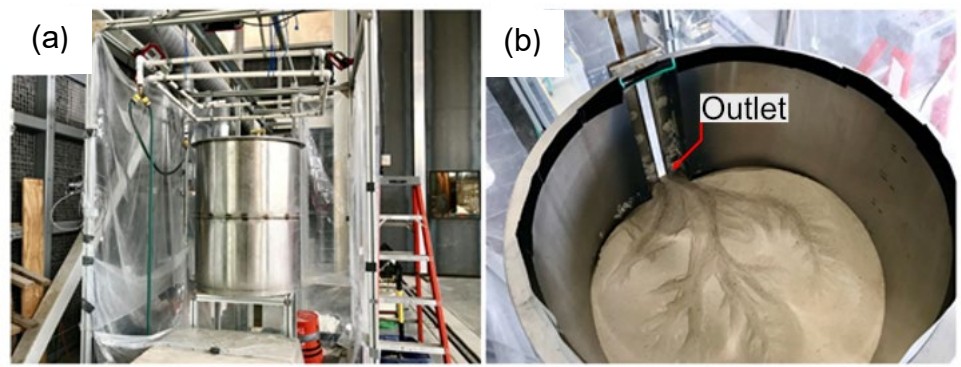

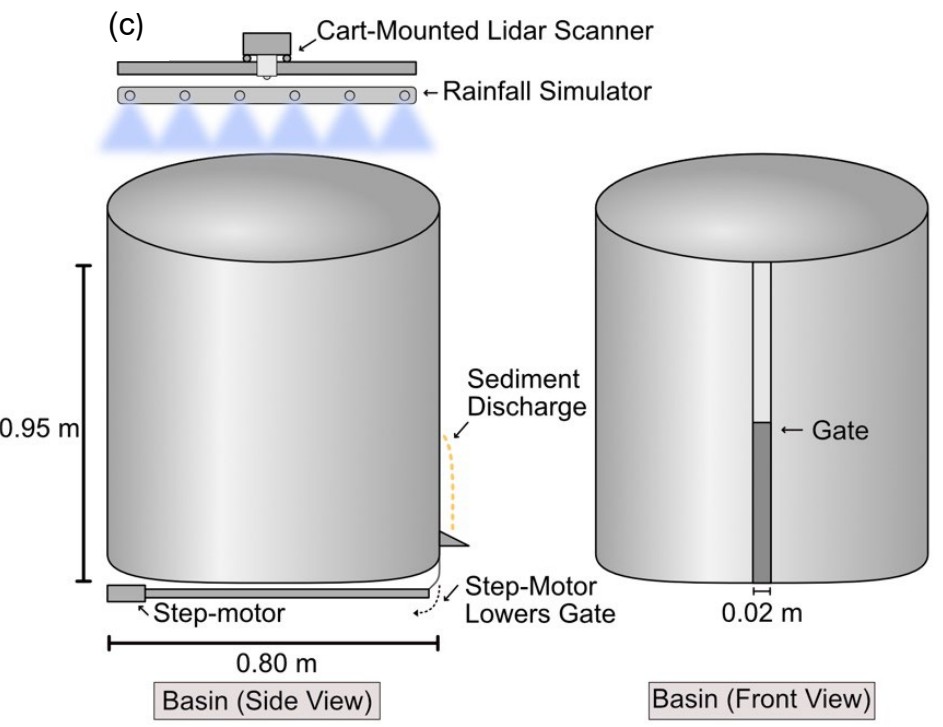

**Figure 2. (a) Image of the basin with the rainfall simulator suspended above it. (b) Image of the basin's interior after channels developed in the substrate. (c) Schematic of the basin with key features labeled.**

The substrate consisted of silica sand ($d_{50}$ = 100 μm) mixed with varying amounts of kaolinite clay. Clay both increases cohesion of the substrate and reduces infiltration capacity [Table 1]. The sand and clay were mixed in a cement mixer and sieved to remove any clumps before adding to the basin. Sediment was added to the basin in 5 cm increments, sprayed with a fine water mist, and compacted by hand with a flat trowel until the sediment package was flat and 25 cm thick. After all the sediment was added to the basin, it was sprayed with a water mist until pooling appeared on the surface, indicating complete saturation of the substrate. By starting with saturated sediment, channel formation by overland flow could begin at the onset of an experiment. Although the initial condition of full saturation biases erosional processes towards overland flow at the beginning of the experiments, it does not impact the partitioning of flow between surface and subsurface later in the experiments once the basin has some relief and flow through both the surface and the subsurface can occur.

We measured infiltration capacity of each sediment composition using a single ring infiltrometer constructed of a 30 cm long cylindrical tube. The tube was placed vertically over a bed of pea gravel to allow for drainage and loaded with sediment to a thickness of 15 cm. After saturating the sediment, water was then added to the tube to a depth (head) of 10 cm. The time needed for the falling head to completely infiltrate the sediment was recorded, allowing the infiltration capacity to be calculated. The test was repeated a several times, and the average value was reported in Table 1. Cohesion was not directly measured. Other experiments that use mixtures with varying amounts of kaolinite mixed with silica sand have found measureable increases in cohesion, yield strength, and critical shear stress with additions of kaolinite clay between 5% and 40% kaolinite. When extrapolated back to 0 – 6% kaolinite clay, it represents an increase of cohesion from 0 to 10 kPa or shear stress from 3 to 7 Pa (Ilstad et al., 2004; Marr et al., 2001; Reddi and Bonala, 1997).

Precipitation was sourced from a set of 20 vegetable misting nozzles suspended 50 cm above the basin on four sides. Precipitation was controlled via a valve outfitted with a gauge that measured water pressure. Pressure was calibrated to specific rainfall rates by measuring the volume of water that fell into the basin over a ten-minute duration. The spatial distribution of rainfall entering the basin varied depending on the nozzle configuration used for an experiment. We measured the spatial variability of rainfall for all nozzle configurations using an array of cups distributed evenly about the basin to measure the volume of water that fell in certain areas. Changing the nozzle configuration was done by covering select nozzles with tape to attain rainfall rates below 16 μm/s while maintaining adequate water pressure for water atomization. A common issue during several experiments was large water droplets contacting the substrate and forming small depressions. This was caused by rainfall coalescing on the cart track above the basin and dripping onto the substrate. The issue was controlled for Run 3 and all subsequent runs using oscillating fans to divert the rainfall away from the tracks. The depressions that do appear in imagery

from later runs occurred only at the start of the experiments while an appropriate fan arrangement was established. Some precipitation collected on the walls of the experimental drum, which could influence channel development along the edges of the experimental basin. To account for this possibility, all drainages along the edge of the basin were removed from digital terrain analyses.

**3.2 Digital Terrain Analysis**

Topographic data collected by the lidar scanner were trimmed to the basin area using FARO® SCENE software. Horizontal and vertical alignments of trimmed scans were assessed and corrected, if necessary, using CloudCompare software. Digital elevation models (DEMs) were generated from point cloud data by performing an inverse distance weighted interpolation for each scan with ArcGIS software. Resulting DEMs had 2 mm x 2 mm raster cells and were edited to eliminate cells that included the basin wall. All further topographic analyses of the DEMs were completed using ArcGIS.

The first goal was to differentiate areas contributing surface water to channels, contributing areas (CAs), from internally-drained surface non-contributing areas (NCAs). A combination of filled DEMs (i.e., small areas of internally-drained cells filled to the local outlet) and unfilled DEMs (i.e. raw data) were used to perform this analysis in four steps [Fig. 3]. First, cells on unfilled DEMs were identified where surface water flow terminates in internally-drained depressions rather than an outlet using the ArcGIS tool "Sink". "Sink" identifies cells that do not drain to the edge of the DEM, which it assumes to be an outlet.

A limitation to this approach is that cells draining to the basin's edge in locations other than the outlet remained unclassified. If internally-drained cells occurred in an area where a channel was visually present, they were assumed to be within the noise of the lidar data and were not considered to be interally-drained. After locating internally-drained cells, their watersheds were delineated by mapping out areas that drained into internally-drained cells (sinks) using the ArcGIS command "Watershed". The areas that drain into internally-drained sinks provide the total NCA as raster cells which were then converted to polygons

[Fig. 3A]. Next, all internally-drained sinks were filled up to their local outlets using the ArcGIS command "Fill". When watersheds were delineated on these filled DEMs, they identified all potential CA to the outlet for the theoretical watershed polygons that would exist if the basin had no NCA [Fig. 3B]. Channels with watersheds that formed along the edge of the basin were eliminated as their formation could be driven by the focused water flow along the basin wall rather than natural processes. Lastly, NCA polygons were removed from the potential CA polygons to provide a final CA for all channels in the

basin [Fig. 3C]. The results from this analysis were sequential scans showing the total surface water CA and NCA in the basin as defined by the topography [Fig. 3].

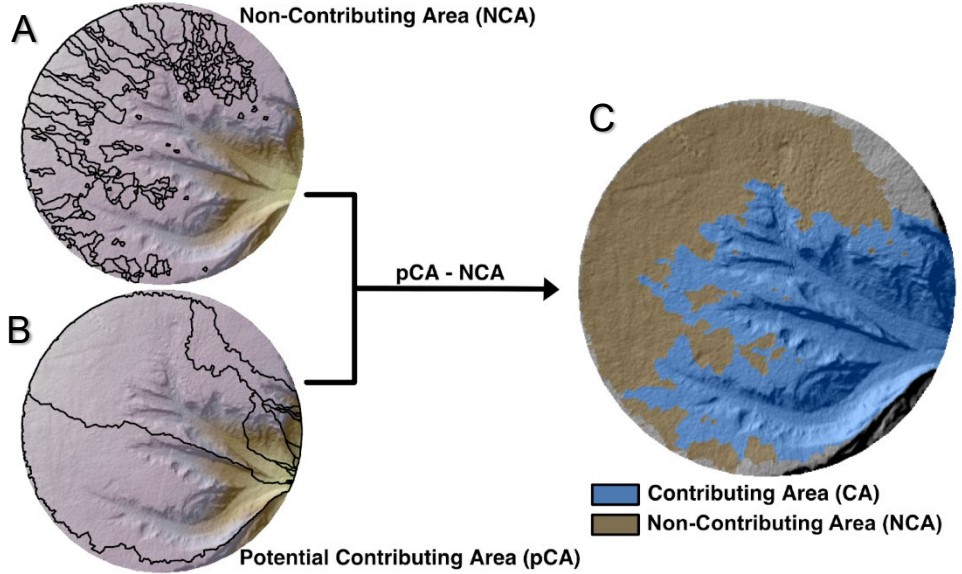

**Figure 3. Process of generating contributing area (CA) and non-contributing area (NCA) from a DEM. (A) Delineation of NCA on the upland surface where each polygon is the watershed of an internally-drained depression. (B) Delineation of potential CA where each polygon is the watershed of a channel, including the internally-drained watersheds (NCA). Note: CA polygons of channels that formed along the edge of the basin were removed. (C) Result of differencing the CA polygons that overlap with NCA polygons to produce a final CA polygon for each watershed. Grey areas represented cells not classified as either CA or NCA.**

The delineated CA included two distinct components: (1) channelized area and (2) non-channelized upland area that supplied surface water to the channel heads. To isolate the two CA components, we first created elevation contour lines using the "Contour" tool on an unfilled DEM with a contour interval of 0.001 m. By selecting contour lines from different elevations and bridging small gaps between segments manually, a single boundary line that outlined the channelized extent of the drainage network was created. This boundary was used to split the CA polygons into "upland CA" and "channelized CA" components at each timestep of a run. Some channel heads were too small-scale for elevation contour lines to capture, but this methodology provided enough precision to determine the area of each component.

During an experiment, NCA was converted to CA as the drainage network expanded. The NCA integration rate is defined as the area of NCA converted to CA per hour. The rate was computed by differencing the area classified as CA in the evaluated timestep from the area classified as NCA in the preceding timestep and dividing by the total time between runs.

The experiments produced two distinct channel head morphologies: Type 1 and Type 2 [Fig. 4]. Type 1 channel heads were
v-shaped with low slope and low relief headwalls. Type 2 channel heads were amphitheater-shaped with high slope and high
relief headwalls. To identify when and where each head type was present, we extracted slope and local relief values from
characteristic channel heads across multiple timesteps and experimental runs. The values were extracted from a 2 mm buffer
around a line drawn along the channel head perimeter. The buffer was directed towards the valley to exclude upland areas.
Slope was calculated within the buffer using the "Slope" tool, while local relief was calculated using the "Focal Statistics"
tool to assesss the elevation range in a three-by-three moving window of cells. These values were used to classify cells
throughout the basin as either "Low Slope & Relief" or "High Slope & Relief," which corresponded to values extracted from
Type 1 and Type 2 channel heads, respectively [Table 3]. Based on the characteristics measured from characteristic channel
heads (Table 3), cells were classified as "Low Slope & Relief" when their slope was 8.4 - 24.7 degrees  and local relief
within the 2mm buffer was 0.0008 - 0.0026 m. Cells were classified as "High Slope & Relief" when they had a slope > 24.7
degrees and relief > 0.0026 cm [Table 3, Fig. 5].

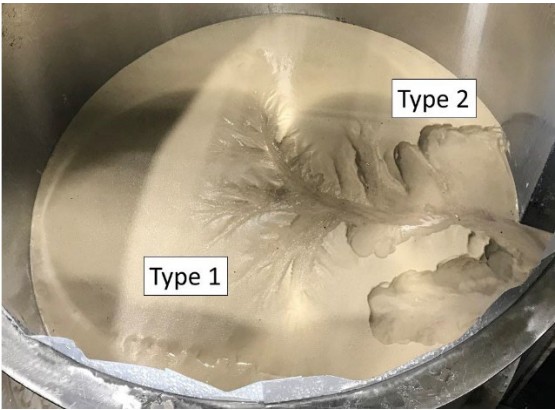

**Figure 4. An example from Run 3 showing the two main kinds of channel heads.  Type 1 had low slope and relief channel heads and Type 2 had high slope and relief channel heads. Image is 0.8 m across from wall to wall.**

**Table 3. Slope and local relief values extracted from Low Slope and Relief (Type 1) and High Slope & Relief (Type 2) channel heads.**

| Statistic | Low Slope & Relief | | High Slope & Relief | |
|---|---|---|---|---|
| - | Relief | Slope | Relief | Slope |
| - | m | degree | m | degree |
| (+1) Std. Dev. | 0.0026 | 24.7 | 0.0085 | 53 |
| Average | 0.0017 | 16.6 | 0.0049 | 25 |
| (-1) Std. Dev. | 0.0008 | 8.4 | 0.0013 | 17.1 |


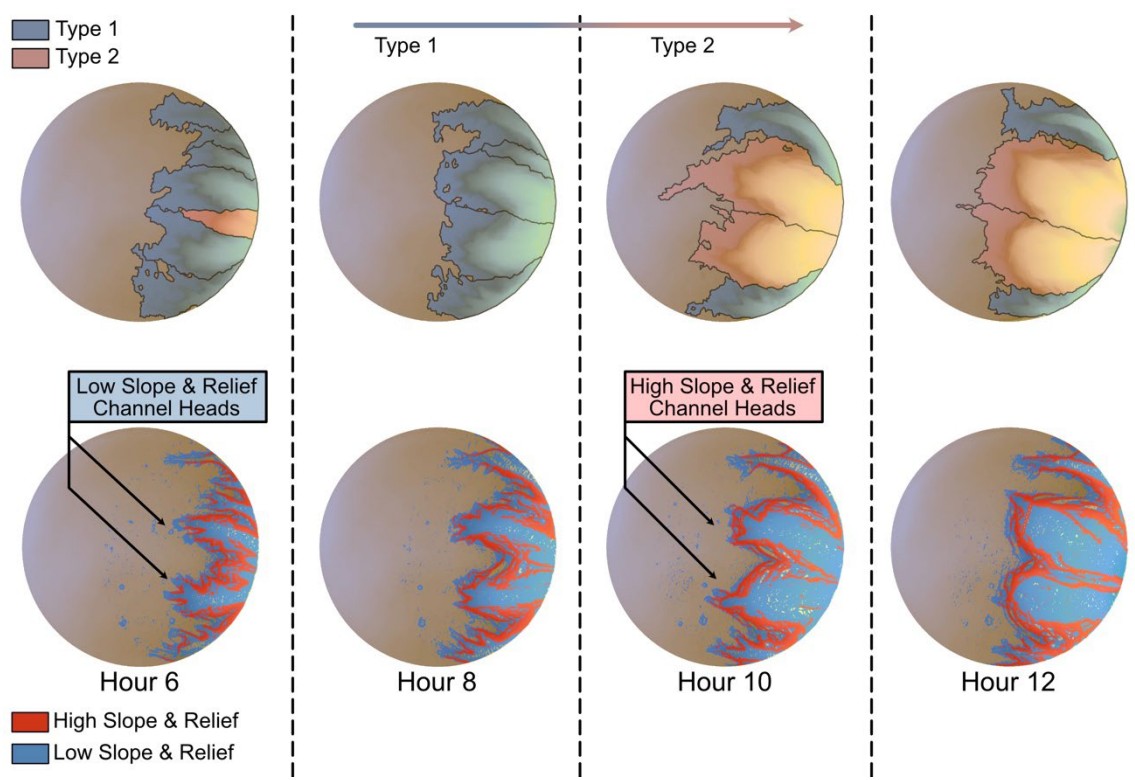

**Figure 5. An example of assigning channel types based on the slope and relief classification of channel heads during Run 3. At hour 6, all but one channel was classified as Type 1 because cells in the vicinity of channel head were classified as "Low Slope & Relief." At hour 8, more cells were classified as "High Slope & Relief" near channel heads, but several cells remained "Low Slope & Relief," maintaining Type 1 classification. From hour 10 onwards, all cells at two central channel heads were classified as "High Slope and Relief," therefore the channels were classified as Type 2.**

By observing the relief and slope classification of cells at channel heads, we assigned a dominant channel type, Type 1 or Type 2, to each sub-watershed (i.e., CA polygon). Channels were classified as Type 1 when cells at the channel head had a "Low Slope & Relief" classification, while channels were classified as Type 2 when cells at the channel head had a "High Slope & Relief" classification. Many channel heads had adjacent cells from both slope and relief categories, which complicated the task of assigning a dominant channel type. Channel heads were classified as Type 2 only when all cells at channel heads had a "High Slope & Relief" classification. Although only slope and relief characteristics were used in classifying channel heads, we found that Type 1 and Type 2 channels tended to have different planview geometry of drainage networks: Type 1 channels often formed branching, dendritic networks while Type 2 channels formed a single wide valley.

With sub-watersheds classified by the dominant channel type, we calculated the incision and volumetric erosion rates associated with each type. To do this, we calculated the depth of sediment eroded by subtracting DEM elevation values from

sequential timesteps and divided by the time between scans to get a local incision rate. To get incision rates for Type 1 vs. Type 2 channels, incision rates for cells in each classification were averaged together. At each cell, the depth of incision was then converted to a volumetric rate of erosion by multiplying the change in depth by the raster cell size of 4 mm². Only the channelized area polygons were used to aggregate the incision rates and volume of sediment removed as non-channelized areas often had small amounts of change (mean of $\approx 0.001$ m) likely caused by both the lidar unit's ranging error of $\pm 0.002$ m and small amounts of sediment diffusion across the upland surface. Any positive values were assumed to be within the lidar data ranging error and set to a value of zero.

## 4. Results

### 4.1 Channel network expansion

Channel network growth resulted in decreasing NCA through time as upland area was captured by the drainage network, converting NCA to CA [Fig. 6]. The initial surface was void of channels, and precipitation was routed to small internally-drained depressions exclusively. Once channel development began, NCA was integrated into the drainage network as channel heads extended into the uplands and breached shallow drainage divides of the depressions. Channels also integrated NCA when their valleys widened via mass wasting. By the conclusion of most experiments, channels had reached their maximum extent and nearly the entire basin surface was CA. In most runs, channel development accelerated at the beginning, remained at a constant rate for a majority of the time, and then slowed near the experiment's conclusion when the basin was near full integration [Fig. 6].

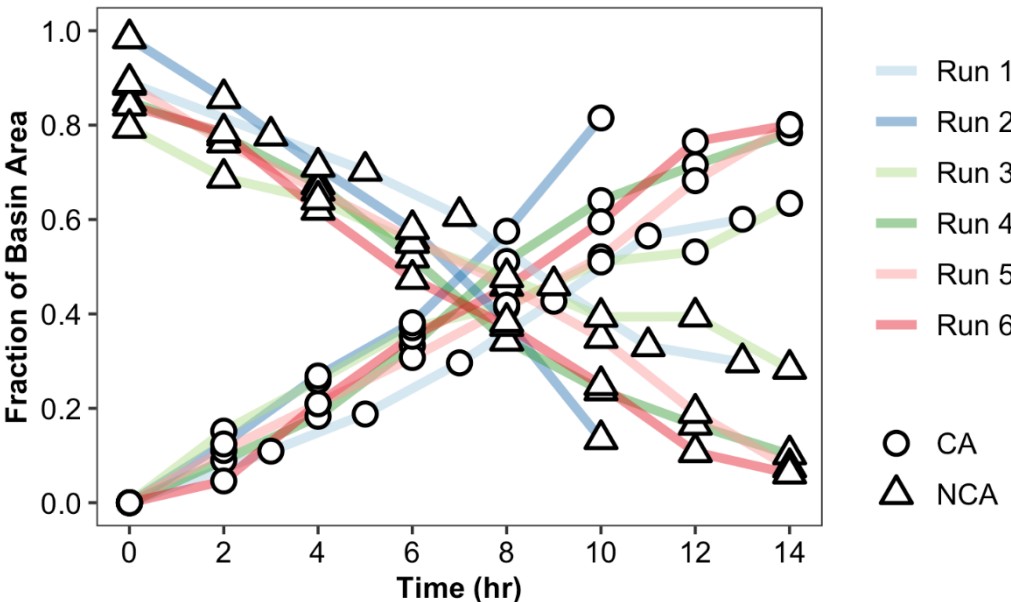

**Figure 6. Time-series of the contributing area (CA) and non-contributing area (NCA) of each experiment. CA increased through time, while NCA decreased for all runs. CA and NCA values are normalized by basin area.**

To investigate the impacts of different experimental conditions, we isolated the effects of a single condition by averaging
multiple experiments with the same substrate, but different rainfall rates (and vice versa) [Table 2]. Run 1 and 2 were excluded from these analyses because they had run durations, scanning intervals, substrate compositions, or rainfall rates that precluded comparison with other experiments in these analyses. All experimental conditions followed a similar temporal pattern and did not produce statistically significant differences in mean NCA integration rate [Fig. 7]. NCA integration rates rose early in the experiments and then slowly decreased, ending at just under 5% of the basin area integrated per hour.

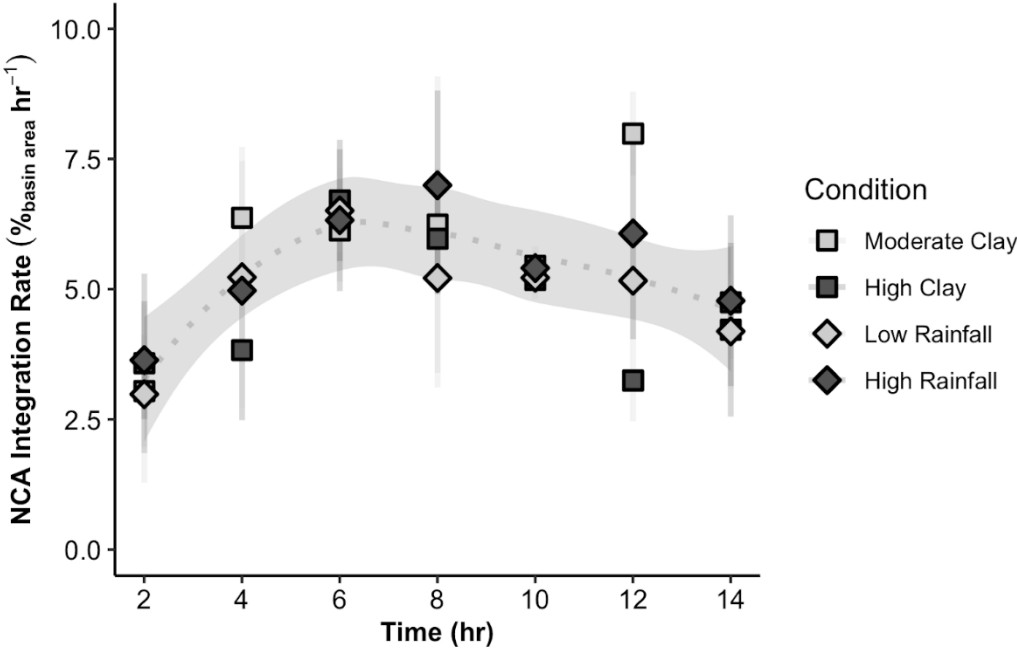

**Figure 7. Average NCA integration rates through time per experimental condition. The rates increased until reaching a maximum at hour six, then declined until the experiment concluded. Error bars are the standard deviation between experiments at equivalent timesteps. The shaded area is the 99% confidence interval of a locally weighted smoothing regression curve of the average values.**

## 4.2 Types of Channel Development

All experiments had two types of coevolving channel development that were differentiated by their morphology. Type 1 channels had dendritic drainage patterns with gently sloping first-order channel heads, while Type 2 channels had single high relief, high slope channel heads (Figure 4). Longitudinal profiles extracted from Type 1 and Type 2 channels show linear profiles in Type 1 channels from mouth to headwaters, while Type 2 channels were concave and steep at the top (Digital Supplement Figure S7). We interpret these as developing from overland flow vs. seepage, respectively, and elaborate on this interpretation in the discussion section. Because the basin started fully saturated with no relief, all runs started with overland flow. Overland flow was visible across the surface of the substrate. System behavior later in the experiments, after relief had developed enough that either surface or subsurface flow could occur, demonstrate how differences in precipitation and substrate impact the processes through which networks expand.

The following descriptions provide a brief overview of key events from each run with an emphasis on channel type classification [See Digital Supplement Figures S1-S6 for imagery of each run]. Run 1 (low clay, moderate rainfall) developed both a single overland flow channel and seepage erosion channel at the experiment's onset. The channels continued to grow throughout the experiment and valley wall widening was extensive after hour 7. Run 2 (moderate clay, high rainfall) channels initially developed via overland flow exclusively. A seepage erosion channel formed at hour 6; however, the channel classification returned to overland flow later in the experiment. Run 3 (moderate clay, low rainfall) channels initially developed via overland flow exclusively. By hour 6, a single seepage erosion channel formed, but was integrated into an overland flow channel due to the collapse of a drainage divide. After hour 8, two overland flow channels transitioned to seepage erosion and maintained this classification for the remainder of the experiment. Run 4 (moderate clay, high rainfall) channels initially developed by overland flow exclusively. At hour 4, a single seepage erosion channel formed, but was integrated into an overland flow channel due to the collapse of a drainage divide between hour 6 and 8. At hour 12, an overland flow channel transitioned to seepage erosion and maintained this classification for the remainder of the experiment. Run 5 (high clay, high rainfall) channels initially developed by overland flow exclusively. At hour 4, a single seepage erosion channel formed that continued to expand for the remainder of the experiment. At hour 12, an overland flow channel transitioned to seepage erosion and maintained this classification for the remainder of the experiment. Run 6 (high clay, low rainfall) channels initially developed by overland flow exclusively. At hour 6, a seepage erosion channel formed and continued to expand for the remainder of the experiment.

From these observations of individual experiments, a few common temporal patterns were noted. At the onset of experiments, overland flow channel heads formed near the basin outlet where a constant base level fall caused channel incision. As knickpoint migration eroded into the basin uplands, channels bifurcated and formed first-order channel heads. After these channels had established a drainage network during the first six to eight hours of an experiment, seepage erosion began to

supersede overland flow. The largest seepage-driven channels formed when mass wasting of channel heads began to increase in frequency and magnitude, causing the channel to attain an amphitheater morphology [Fig. 8]. Drainage divide collapse and channel coalescing eliminated some of the smaller seepage channels, while the larger channels often persisted until the conclusion of the experiment.

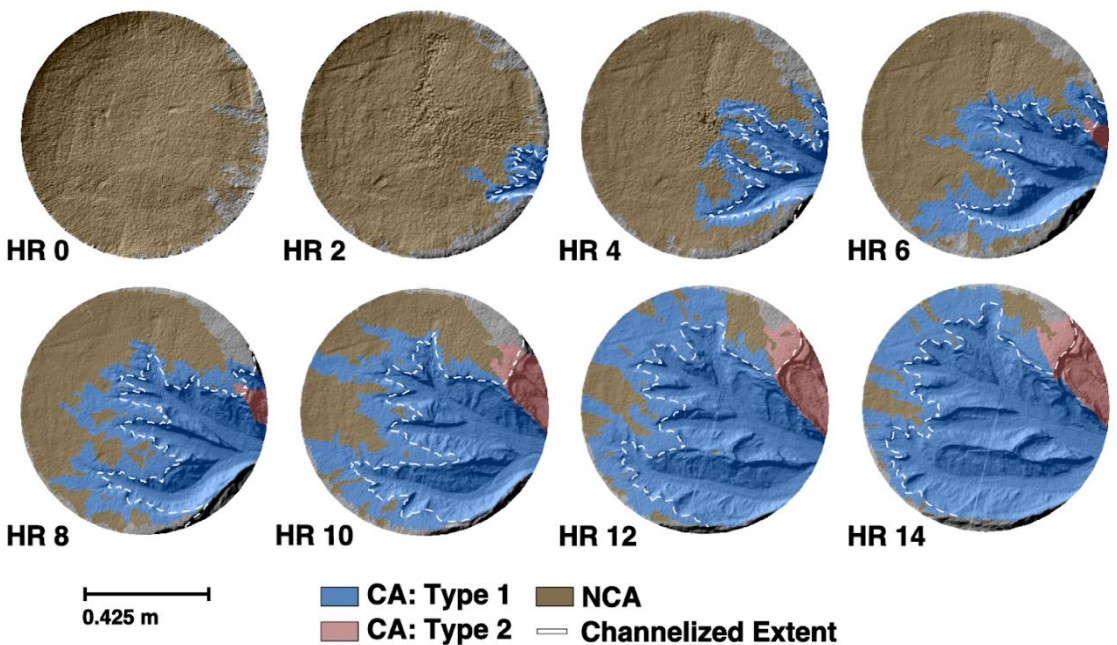

**Figure 8. Example of classifying CA polygons as Type 1 from overland flow erosion (blue) or Type 2 from seepage erosion (red) at each timestep for an experiment (Run 6). Overland flow channels developed exclusively until hour 6 when a seepage erosion channel began to form. The channelized extent (white dashed line) approximately separates the CA polygons into upland CA and channelized area components. Refer to the Digital Supplement for imagery of other runs.**

Each experiment had varying amounts of each type of channelization over its duration. Overland flow channels, on average, comprised a majority of total channelized area for most experiments and were dominant during the first half of all experiments. After hour 8, some experiments had a sharp decrease in channelized area from overland flow after these channels transitioned to seepage channels between timesteps [Fig. 9]. The largest of such decreases occurred during Run 3, which was the only experiment where seepage channels obtained a majority of the channelized area. Run 6 was notable for maintaining the largest average fraction of overland flow-driven channels, with 95% of total channelized area, over the experiment's duration. Run 1

was the only experiment where seepage erosion began at the experiment's onset and channelized a substantial area before hour 8.

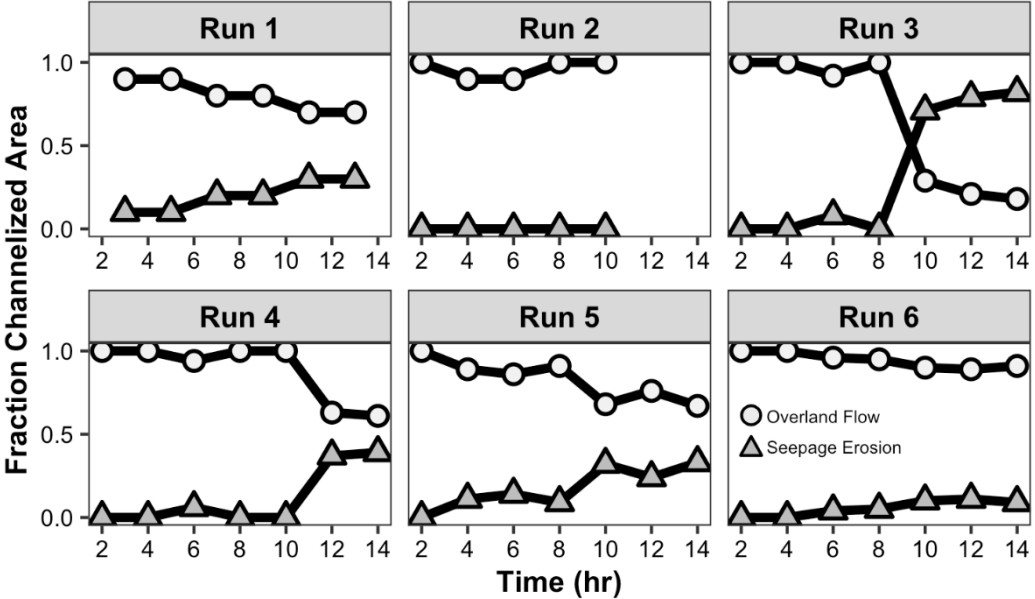

Figure 9. The fraction of total channelized area with channel head erosion driven by overland flow and seepage erosion through time. The fraction of channelized area occupied by seepage erosion increased after hour 8 in most experiments. Channelized area values are normalized by the total channelized area.

Analysis of the results by parameter showed the consistency of type of channel development was affected by both the substrate composition and rainfall rate [Fig. 10]. High clay experiments had a greater and more consistent amount of overland flow channelization compared to moderate clay experiments. For high clay experiments, channelized area from overland flow increased linearly through time at a rate of 0.03 $m^2\,m^{-2}\,h^{-1}$ and reached a maximum of 0.42 ± 0.05 $m^2\,m^{-2}$ by hour 14. Areas reported are channelized areas ($m^2$) normalized by total basin area ($m^2$). The average standard deviation of overland flow channelized area for high clay experiments was 0.02 $m^2\,m^{-2}$, four times less than moderate clay experiments. Seepage channels proliferated under moderate clay conditions, with most growth in channelized area from seepage erosion coming after hour 8.

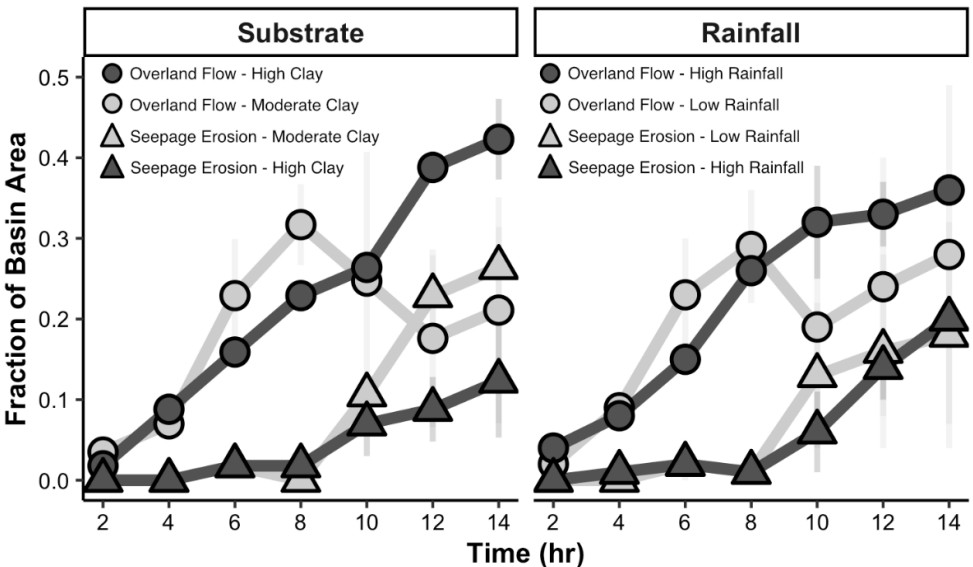

**Figure 10. Average channelized area of overland flow and seepage channels normalized by total basin area through time for experiments with moderate or high clay substrate (left) and low or high rainfall (right). After hour 8, seepage erosion channelized area increased to a greater extent under a moderate clay substrate than high clay substrate. High clay substrates and high rainfall rates resulted in greater overland flow channelized area. Error bars are the standard deviation in the channelized area between experiments at each timestep. Channelized area values are normalized by total basin surface area.**

On average, high rainfall rates resulted in greater and more consistent channelized areas of both channel types, but particularly for channel heads eroding through overland flow [Fig. 10]. High rainfall rate experiments led to a linear increase in channelization by overland flow through time, reaching a maximum normalized channelized area of $0.36 \pm 0.02$ m$^2$ m$^{-2}$ at hour

14. The average standard deviation of channelized area by overland flow in the high rainfall runs, 0.03 m$^2$ m$^{-2}$, was three times less than low rainfall rate experiments. Low rainfall rate experiments also differed in that channelized area from overland flow decreased between hour 8 and 10 as more channels transitioned to seepage channels. However, unlike the moderate clay experiments, the decrease was not as sustained; channelization by overland flow continued to increase from hour 10 onwards.

Rainfall rates appeared to have less of an influence on the total area of seepage channelization. The main difference was the variability in channelized area: high rainfall rates had an average standard deviation of 0.02 m$^2$ m$^{-2}$ compared to 0.05 m$^2$ m$^{-2}$ for low rainfall rates.

Temporal changes in upland CA that supplied water to channel heads via surface flow followed a similar pattern to NCA integration rate through time [Fig. 7, 11], rising initially, then decreasing slowly over the remainder of the experiment. Under

all conditions, overland flow channels had a greater average upland CA, 0.11 m² m⁻², compared to seepage channels, 0.02 m²
      m⁻². The large standard deviations around hours 8 to 10 correspond with the onset of seepage erosion in many experiments.
      Seepage channels that formed by transitioning from overland flow channels under moderate clay and low rainfall conditions
      accounted for a majority of the variance.

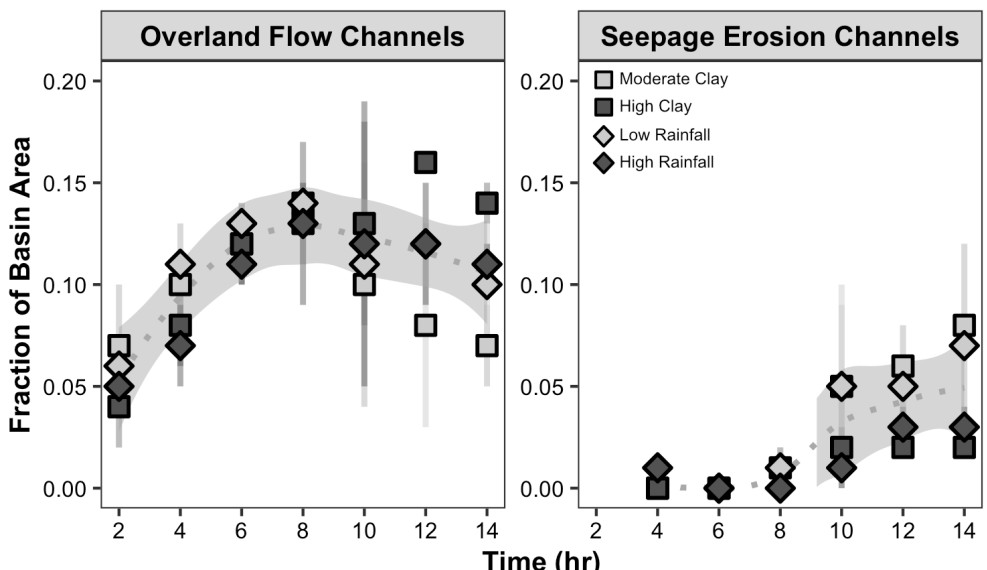

**Figure 11. Upland CA of overland flow (left) and seepage (right) channels through time for different experimental conditions. Upland
      CA for overland flow channels increased until reaching a maximum at hour 8, then declined until the experiment concluded. Upland
      CA for seepage erosion channels increased after hour 8 but were smaller compared to overland flow,  Averages at each time step
      are plotted, with error bars indicating the standard deviation between experiments. The shaded area is the 99% confidence interval
      of a locally weighted smoothing regression curve of the average values. Contributing area values were normalized by the total basin
surface area.**

## 4.3 Erosion

Channel networks expanded, eroding sediment from an increasing fraction of the basin through time. Areas in the basin prone
to erosion include channel heads, valley floors, and valley walls. Low magnitude erosion occurred along the valley floor, where
the erosion depth between timesteps was on the order of one to two centimeters. The highest magnitude of erosion occurred at
valley walls and drainage divides by mass wasting, which could remove multiple centimeters of sediment in a single event
[Fig. 12].

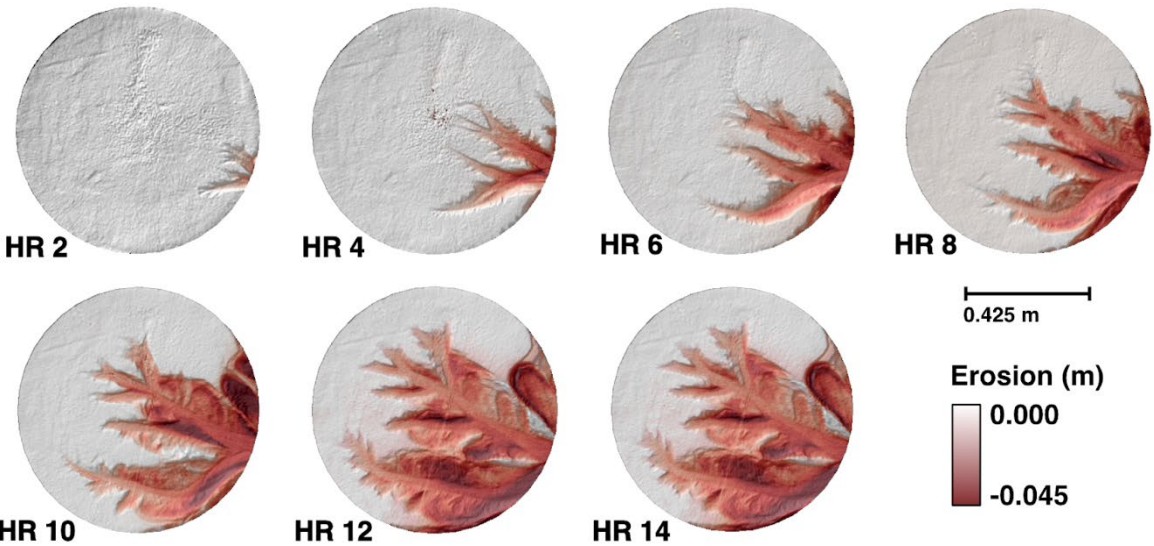

**Figure 12. DEM time-series of erosion depth per timestep throughout Run 5. Erosion depths were greatest along valley walls and drainage divides where mass wasting was most common. Red (negative) indicates erosion.**

Erosion volumes increased through time for all experiments; however, the total erosion volume differed between experiments. We assessed the erosion of each channel type independent of duration and channelized area by calculating incision rates (erosion volume divided by the channelized area per time) and then normalizing by the rate of base level fall. Erosion rates that perfectly match the rate of base level fall would be equal to 1. For all experiments, average incision rates of seepage channels were greater than overland flow channels [Fig. 13].

The incision rate of overland flow channels increased during the early period of channel establishment, then equilibrated at a value about half of the rate of base level fall, 0.46 on average. An exception to this pattern of equilibration was Run 5 which had a drainage divide collapse between hours 12 and 14 causing a sharp rise in incision rate. Seepage channels had fewer incision rate observations than overland flow channels because they formed later in the runs and were sometimes eliminated by drainage divide collapse. The incision rates associated with seepage channels were closer to the rate of base level fall, 0.83 on average. Run 1 had an exceptionally high incision rate for seepage channels, averaging 0.98, a value greater than all other experiments and nearly equivalent to the rate of base level fall.

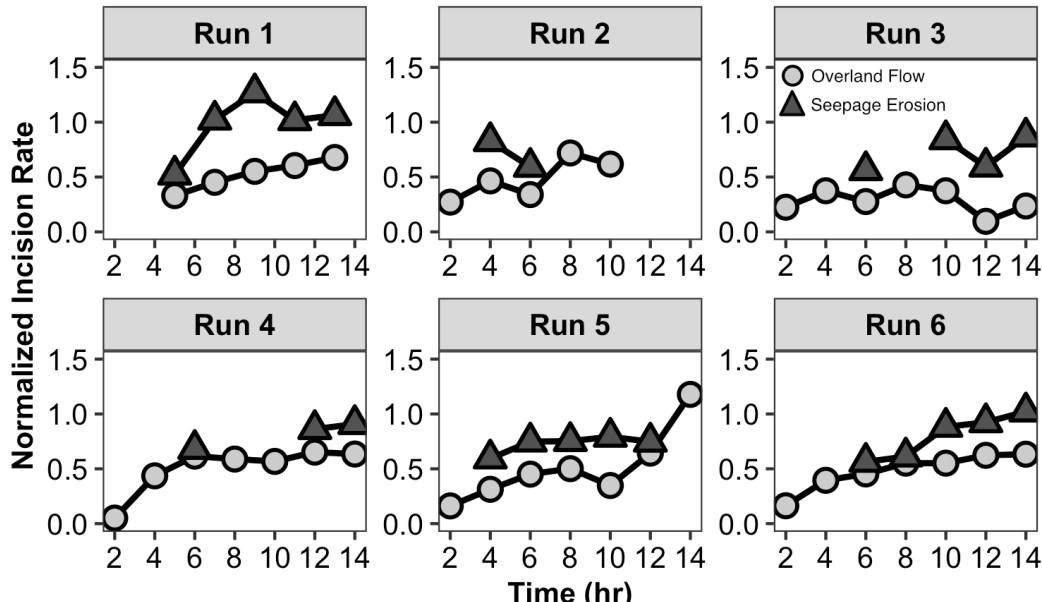

**Figure 13.** Normalized incision rates of overland flow (circles) and seepage (triangles) channels throughout each experiment. Incision rates were greater for seepage erosion channels then overland flow channels during all runs. A normalized incision rate of 1 indicates that the incision rate equals the rate of base level fall.

In terms of the total erosion, experiments with conditions that led to greater amounts of channelization, high clay and high rainfall [Fig. 10], eroded at higher rates and in larger volumes than other conditions [Fig. 14]. High clay experiments had more erosion primarily from overland flow channelization [Fig. 10], while high rainfall experiments had a greater contribution from both types of channels. Under conditions with more channelization from seepage, such as moderate clay and low rainfall [Fig. 10], seepage channels eroded similar or greater volumes of sediment than overland flow channels. In all cases, seepage channels were only a substantial erosion source after hour 8. The sudden rise in erosion volumes between hour 12 and 14 for high clay and rainfall experiments was due to the drainage divide collapse during Run 5. However, even when Run 5 was excluded from the final timestep average, the high clay and rainfall experiments still maintained greater erosion rates and volumes than the other conditions.

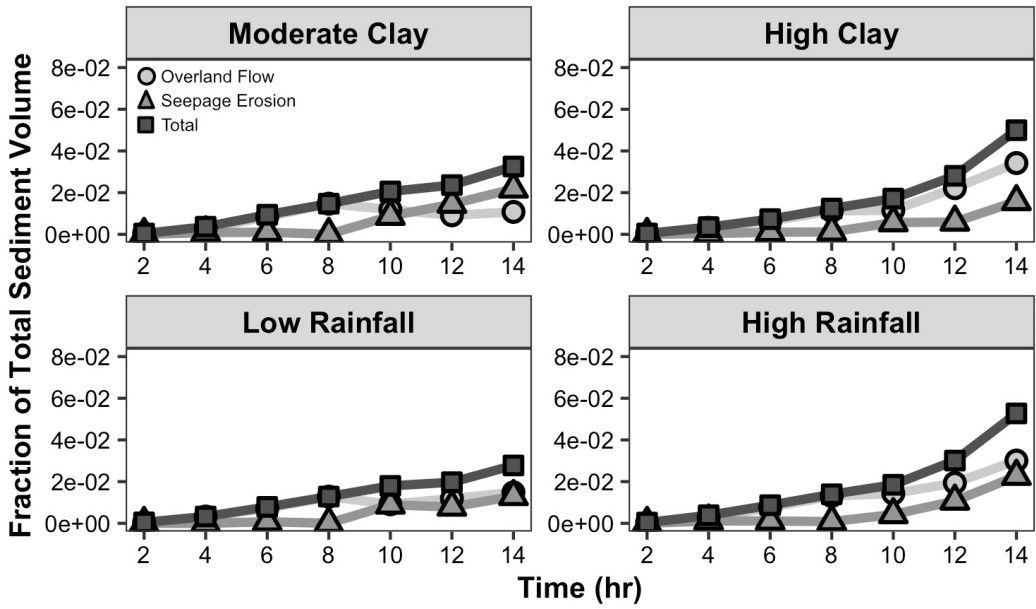

**Figure 14.** Average erosion volume of overland flow and seepage channels under different experimental conditions per timestep. Seepage erosion produced greater erosion volumes under moderate clay and low rainfall conditions compared to overland flow erosion, which dominated in high clay and high rainfall conditions. The total erosion volume is the sum of both erosion values per timestep and was higher in high clay and high rainfall conditions. Error bars are the standard deviation in erosion volume between experiments and are mostly smaller than the data symbols.

## 5. Discussion

The experiments described here focus on drainage evolution on a low-gradient surface subjected to a constant supply of rainfall and base level fall. These conditions are not unique to our experiments. Base level changes are common in post-glacial environments, associated with processes like glacial lake drainage, valley incision from high discharge glacial meltwater events, or differential uplift associated with isostatic rebound. Although many of these incisional triggers are abrupt, the upper watershed experiences base level fall as a more continuous process as incision propagates upstream, similar to the experiments here. For example, incision of the Minnesota River valley by glacial meltwater has led to progressive and on-going drainage extension and incision by tributaries into relatively flat-lying glacial tills and glaciolacustrine sediments in southern Minnesota (Gran et al., 2009, 2013) [Fig. 15]. Likewise, drainage of major glacial lakes like glacial Lake Duluth lowered base level to streams draining into Lake Superior by over 200 meters (Grimaud et al., 2016), leading to incision that continues to migrate upstream over time into glacial tills and glaciolacustrine sediments [Fig. 15]. Incisional waves associated with base level fall

are driving network extension in similar watersheds across large swaths of the Central Lowlands, and the experimental results here give additional insight into the processes driving network expansion and which conditions favor overland flow vs. seepage

erosion.

Although field examples highlight areas where drainage network expansion is occurring through similar processes of overland flow and seepage erosion, the model was not designed to be a scale model of those areas. Instead, it was designed to be a process model, to demonstrate whether varying conditions of rainfall and substrate could lead to different processes of channel network growth and development. Previous experiments have focused predominantly on providing ideal conditions for either

surface water or groundwater-driven processes of channel development (Berhanu et al., 2012; Bonnet and Crave, 2003; Hasbargen and Paola, 2000; Lague et al., 2003; Lobkovsky et al., 2004; Parker, 1977; Pelletier, 2003; Phillips and Schumm, 1987; Schorghofer et al., 2004; Singh et al., 2015; Sweeney et al., 2015). Our experiments sought to provide a middle-ground with suitable conditions for either process to occur, uniquely demonstrating how both processes could co-evolve in the same low-gradient drainage basin. Temporally, the modelling framework here accelerates network growth by running constantly at

high flow conditions. This is a common tool used in physical (and numerical) modelling to accelerate system evolution, focusing only on the times when erosion happens. Spatially, they lack subsurface heterogeneities, differential strength driven by vegetation, and erosion driven by processes other then precipitation and sapping. However, the model does provide a system with conditions that allow for both surface runoff-driven and subsurface-driven channel head erosion, giving us the ability to better understand what conditions may drive more surface vs. more subsurface erosion in low-gradient, incising landscapes,

and how the dominance of one process vs. the other may vary over space and time.

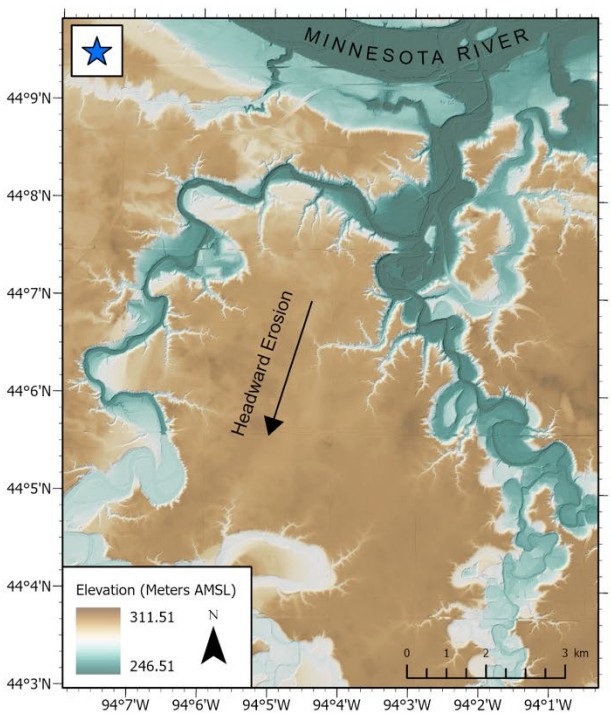 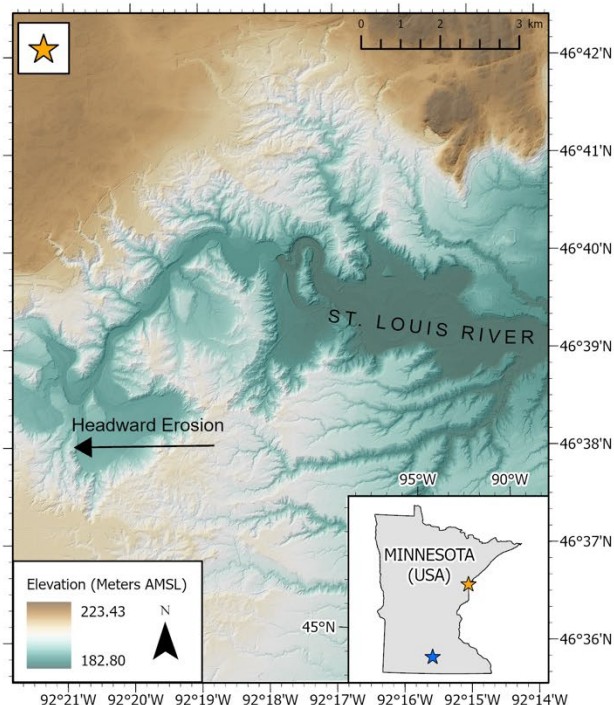

**Figure 15. DEMs of Minnesota River tributaries in south-central Minnesota and tributaries to the St. Louis River in northeastern Minnesota, USA. In both cases, tributaries incised after a base level drop at their outlet and continue to erode headward into the surrounding uplands.**

## 5.1 Channel Development Processes

Our experiments demonstrate that channel development driven by relative base level fall can produce two distinct and coevolving channel types that differ in their morphology and hydrologic characteristics. We attributed these differences to be the result of separate channel-forming processes. Type 1 channels, characterized by large upland CAs and low relief channel heads, we interpret as formed by overland flow where surface water accumulated as it moved downslope and exerted shear stresses high enough to erode the substrate. Overland flow was visible across the upland surface in the experiments. It was an active process early on, in part driven by the initial conditions of the experiments with a fully saturated landscape and no relief [Fig. 9]. As upland CA increased through time [Fig. 11], incision rates from overland flow increased [Fig. 13]. The large upland CA supported numerous first-order channels, creating dendritic drainage patterns with slowly increasing total erosion volumes, as observed in other experiments focused on overland flow channel development (Hasbargen and Paola, 2000;

Parker, 1977; Pelletier, 2003) [Fig. 14]. Although dendritic drainage patterns with long, narrow channels have emerged in experiments that only have seepage erosion (Lobkovsky et al., 2004; Smith et al., 2008), the morphology of those channel heads were more rounded than the Type 1 channel heads.

Type 2 channels formed by seepage erosion where groundwater exfiltrated through channel headwalls with enough force to entrain sediment and cause mass wasting by undermining headwalls. Seepage-driven mass wastingwas rare early on in the experiments [Fig. 10], in part due to the initial conditions of a fully saturated system and in part due to the lack of relief. As relief increased, seepage erosion began. The intermittent nature of mass wasting events caused headward erosion of channels to proceed as large, sporadic failures unlike the consistent cadence of headward erosion by overland flow. The small upland CA of seepage erosion channels [Fig.11] supplied less surface water to channel heads, hindering overland flow. However, 530 subsurface water is unconstrained by surface water divides, and small upland CA does not impact the ability of these channels to draw in subsurface water, allowing seepage erosion to initiate mass wasting which formed high slope and relief headwalls [Table 3]. Both experiments (Berhanu et al., 2012; Gomez and Mullen, 1992; Howard and Iii, 1988; Howard and McLane, 1988; Lobkovsky et al., 2004; Schorghofer et al., 2004) and studies in natural landscapes (Abotalib et al., 2016; Kochel and Piper, 1986; Laity and Malin, 1985; Schumm et al., 1995) have identified amphitheater-shaped headwalls as a common, though 535 not exclusive (Petroff et al., 2011), feature of seepage erosion.

## 5.2 Process Drivers: Substrate, Precipitation, Relief

The degree to which drainage networks develop by overland flow or seepage erosion depends on a number of factors including substrate, rainfall rate, and relief. Field studies in unconsolidated sands and gravels have found that groundwater seepage can play an important role in channel development and formation (Coelho Netto et al., 1988; Dunne, 1990; Lapotre and Lamb, 540 2018; Micallef et al., 2021; Pillans, 1985; Schumm et al., 1995; Schumm and Phillips, 1986; Uchupi and Oldale, 1994). As grain size decreases to silt and clay size fractions, low permeability limits infiltration, decreasing the likelihood of seepage erosion and sapping (Lapotre and Lamb, 2018). Results from our experimental runs are compatible with these field observations; the degree to which drainage networks developed by overland flow or seepage erosion varied as a function of substrate composition [Fig. 10, 11]. Infiltration tests found that differences between the high clay and low clay experiments 545 [Table 1] effectively straddled conditions for seepage erosion feasibility as laid out in Lapotre and Lamb (2018), with high clay experiments approaching conditions where seepage was not possible, while low clay experiments still allowed for seepage erosion to occur. Experiments with a low or moderate clay substrate had both larger infiltration capacities that allowed more water to infiltrate into the subsurface and lower cohesion, making it easier for seepage erosion to occur [Table 1].

The connection between erosion process and rainfall rate is more complicated. In order for channel heads to erode by seepage 550 erosion, there must be enough precipitation that the infiltrating fraction can provide the discharge needed to overcome cohesion holding particles in place. Field studies by Micallef et al. (2021) in a series of coastal gullies in New Zealand, for example,

found a rainfall threshold of 40 mm/day necessary for seepage erosion to occur at that location. Experimental studies find that the velocity of exfiltrating groundwater also must be high enough to remove the eroded particles deposited at the base of slopes, setting up the headwall for continued erosion (Abrams et al., 2009; Howard and McLane, 1988; Onda, 1994). For erosion by overland flow, there must be enough discharge on the surface to generate a high enough shear stress for particles to be eroded and transported downstream. Thus, high precipitation and high contributing area both contribute to greater erosion via overland flow. High rainfall rates coupled with high clay contents had the highest erosion volumes overall [Fig. 14] and were particularly amenable to overland flow over seepage erosion as less of the precipitation that landed on the surface was lost to infiltration. Unlike Berhanu (2012), both elongated, single channels and wide, bifurcated channels formed by seepage erosion under uniform rainfall, suggesting that groundwater flow was influenced by other factors like the presence of adjacent channels or the model boundary to maintain uniform flow to channel heads (Cohen et al., 2015) [Refer to Digital Supplement for additional imagery].

In addition to substrate composition and rainfall rate, relief generated by channel incision was an important control on seepage erosion. During the initiation and early expansion of the drainage network, relief was limited, and only a few channels formed by seepage erosion as channel heads competed for upland CA [Fig. 11]. The dominant channels captured enough upland CA to evolve by overland flow, while the subordinate channels were starved of upland CA and could only grow by seepage erosion. These early seepage erosion channels were often eliminated through time as mass wasting breached small drainage divides, and seepage channels were incorporated into larger channel networks. Later in the experiments, some channel heads became starved of upland CA, and existing channels underwent a process transition from erosion via overland flow to seepage erosion [Fig. 5, 8, 9]. The earliest evidence of process transitioning appeared around hour 6 of most experiments. During this time, the morphology of some overland flow channels began acquiring an amphitheater shape as the frequency of headwall mass wasting increased [Fig. 5]. The resulting amphitheater shape had a smaller upland CA. Substantial amounts of mass wasting occurred after hour 8 when seepage erosion could consistently undermine headwalls [Fig. 9, 14]. By that time, all experiments had experienced 9.2 cm of total base level fall, and relief throughout the basin had increased as channel incision progressed in unison. The greater relief likely exceeded a critical slope stability threshold, and seepage erosion was capable of initiating mass wasting like the experiments by Lobkovsky (2004) demonstrated. Pelletier (2003) similarly noted that as relief increased during their experiments, base flow (i.e., groundwater) became an increasingly important driver of channel growth.

## 5.3 Network expansion and erosion over time

In an evolving post-glacial landscape, NCA extent starts high and declines through time as channel networks evolve and drainage density increases (McDanel et al., in prep.; Meghani et al., in prep.). In our experiments, NCA integration rates did not differ significantly between experimental conditions [Fig. 7] despite causing varying amounts of overland flow and seepage erosion channelization [Fig. 9]. Part of this was related to the dominance of overland flow overall and particularly during the first half of the experiments, when relief was low. In Run 3, where seepage erosion was the dominant erosional process for

multiple hours, NCA integration slowed, which indicates that channel network expansion could slow over time as declining CA and increasing relief allow for more seepage erosion to occur [Fig. 6 – Run 3].

In terms of erosion, overland flow accounted for the majority of the erosional volume in most experiments [Fig. 14], but the rate of incision was higher in areas eroding through seepage erosion [Fig. 13]. Overland flow channels were characterized by slow, consistent erosion of valley floors through time with occasional mass wasting of steep valley walls [Fig. 12]. Individual channels had limited erosional power and incised through the substrate at an average rate about half the rate of base level fall [Fig. 13]. The incision rate did not differ substantially between experiments [Fig. 13]. Overland flow accounted for the majority of erosional volume in most experiments because it channelized larger areas [Fig. 9] that cumulatively eroded more sediment [Fig. 14]. It followed that conditions favoring overland flow channelization – high clay and high rainfall - were associated with the largest erosion volumes [Fig. 10 & 14].

Seepage erosion caused mass wasting, which often removed multiple centimeters of headwall sediment in a single event. The large magnitude of erosion by mass wasting resulted in incision rates greater than overland flow that nearly kept pace with the rate of base level fall [Fig. 13]. In runs with conditions that favored seepage erosion, erosional volumes from seepage were similar to volumes eroded via overland flow in the latter half of the experiments, when seepage erosion was more dominant [Fig. 14]. Run 1 had particularly high incision rates; the less cohesive substrate in Run 1 increased the effectiveness of seepage erosion relative to other experiments [Fig. 13]. However, seepage erosion caused mass wasting episodically over a smaller area for most experiments compared to the area eroded through overland flow, which limited the total volume of sediment eroded by seepage [Fig. 14]. Mass wasting was episodic because of the time needed for exfiltrating groundwater to undermine channel headwalls. Like the experiments by Howard and McLane (1988), numerical modeling by Abrams et al. (2009), and observations in the field by Onda (1994), sediment deposited at the base of headwalls after mass wasting had to be removed for erosion to proceed. Deposits that are not removed can temporarily stabilize slopes until fluvial transport or overland flow removes them, which can slow the rate of channel evolution.

**5.4 Implications for Drainage Network Development**

Different models of drainage network development (e.g., top-down versus bottom-up) can explain how hydrologically-disconnected areas, NCAs, are gradually integrated into the drainage network over time. The experiments presented here explored processes associated primarily with a bottom-up model of drainage network integration driven by relative base level fall, which has been associated with low-gradient settings like plateaus (Whipple et al., 2017), tidal marshes (D'Alpaos et al., 2005, 2007; Fagherazzi et al., 2012), and formerly glaciated landscapes across the Central Lowlands (Gran et al., 2009, 2013). The experiments demonstrate that drainage network development driven by base level fall could proceed by different processes depending on the substrate composition, rainfall intensity, and relief generated by channel incision.

A critical finding was that the dominant process of channel development can transition from overland flow to seepage erosion as channel incision creates more relief over time [Fig. 5, 9, 10]. The degree of channel incision depends on both the magnitude of base level drop and the amount of time incision has had to propagate through the drainage network. Newly developing rivers may not exceed the relief threshold for seepage erosion to consistently cause mass wasting, restricting headward erosion to overland flow. More incised rivers may have generated enough relief to become susceptible to routine mass wasting by seepage erosion, changing the dominant process of headward erosion. The onset of seepage erosion may be particularly important when considering the susceptibility of a landscape to gullying, which seepage erosion can drive and which is a major source of land degradation in many low-gradient settings used as agricultural land (Castillo and Gómez, 2016; Poesen et al., 2003; Valentin et al., 2005). Seepage erosion also allows network expansion to continue even if upland CA is too low for overland flow to exceed erosional thresholds at the channel heads. This could be quite important in disconnected post-glacial landscapes with significant areas in NCA.

The processes of channel development could also affect the pace of NCA integration in low-gradient landscapes. Our experiments suggest if conditions support erosion by overland flow, then channels may integrate NCA at a consistent rate after an early period of channel initiation [Fig. 6, 7]. Channels developing by seepage erosion tend to integrate less NCA, which could reduce the overall rate of NCA integration depending on the pervasiveness of seepage erosion within a drainage basin [Fig. 6 – Run 3]. However, even though seepage erosion integrated NCA at a slower rate, incision rates were higher from seepage erosion, and volumetric erosion rates can be similar under both processes [Fig. 13, 14].

Our analysis assumed that precipitation routed from a topographically-defined CA drove channel development. This assumption might be incorrect if NCA depressions filled with water and overtopped their drainage divides. In addition, CA only applied to surface water contribution, not subsurface, and groundwater from NCA likely crossed surface divides to reach channels during the experiments. Lai and Anders (2018) demonstrated that such hydrologic connections between NCA and drainage networks are critical in driving channel development in low-gradient post-glacial landscapes. While the experiments did not account for NCA surface connections, they underscored how the hydrologic pathway by which potential connections occurred could influence the processes of channel development and the resulting channel morphology.

In light of these findings, we have focused on the implications for drainage networks in the glaciated Central Lowlands region, USA, that developed in a largely low-gradient setting with different glacial deposits, climate regimes, and degrees of channel incision. One example of a system that appears to be expanding via both overland flow and seepage-driven erosion is Mission Creek, a tributary of the St. Louis River in northeast Minnesota, USA [Fig. 16]. Mission Creek is incising into glacial till, glaciolacustrine sediments, and sandstone bedrock following base level fall associated with the drainage of glacial Lake Duluth at the end of the last glaciation (Grimaud et al., 2016). Proximal to the outlet, the channel has higher relief and higher infiltration capacities in the thick sandy to clayey glaciolacustrine near-shore deposits (Lusardi et al., 2019). Upstream, those deposits transition into clay-rich glacial tills and there is less relief overall in the system. The combination of high relief, less cohesive,

and more permeable sediment in the lower watershed favors mass wasting induced by seepage erosion, and many channels located in the lower watershed have amphitheater-shaped headwalls indicative of seepage erosion and mass wasting [Fig. 16, left]. Channels in the upper watershed have channel tips more characteristic of overland flow [Fig. 16, right]. While Mission Creek is not necessarily prototypical of post-glacial rivers elsewhere in the region, it is an illustrative example of how overland flow and seepage erosion may operate within a watershed simultaneously, affecting the processes by which drainage networks expand.


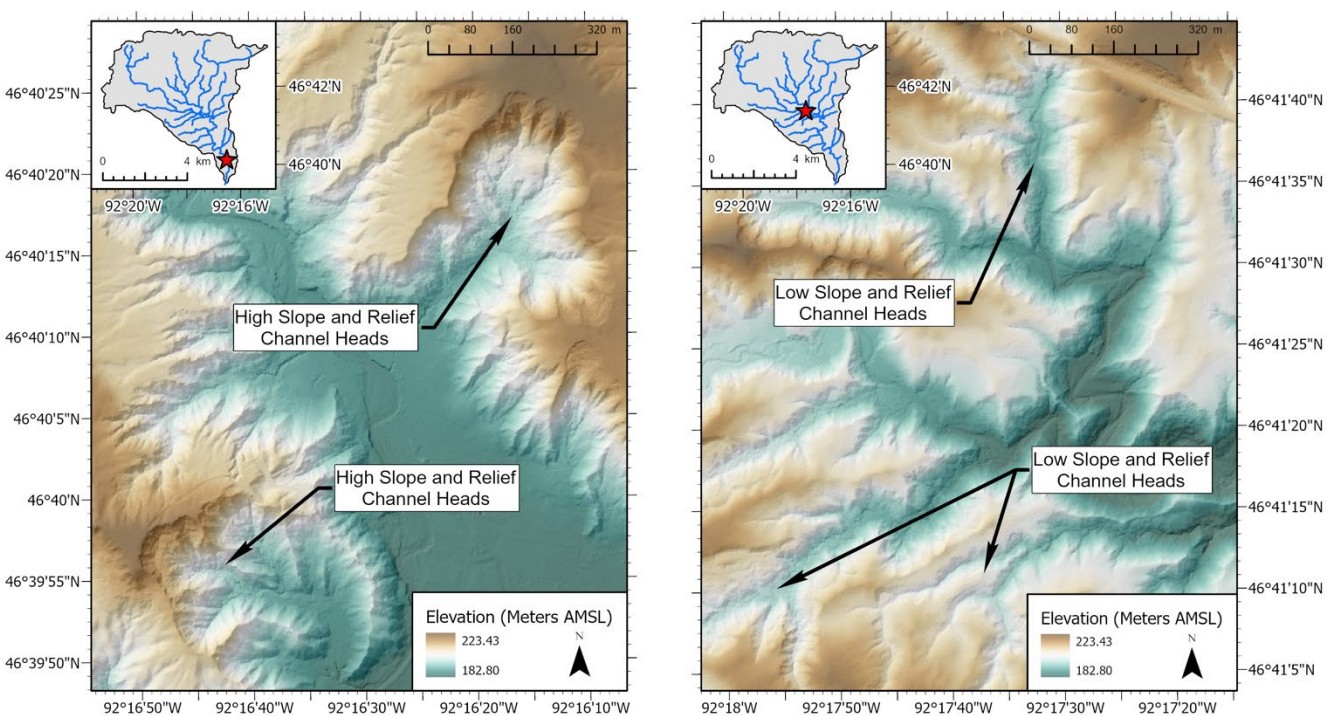

**Figure 16. Comparison of channels in the lower (left) and upper portion (right) of the Mission Creek watershed in Fond du Lac, MN. High slope and relief, ampitheater-shaped channel heads indicative of seepage erosion are more prominent in the lower portion compared to the upper portion,**


The experiments also showed that substrate composition and precipitation rate can influence the processes of channel development [Fig. 10]. The substrate's texture influences infiltration capacities, which can affect whether precipitation is routed to channels by the surface or subsurface, supporting different processes of channel development. As channels incise

through multiple units of glacial sediment, different material properties can introduce complex relationships between precipitation routing and channelization, making it difficult to predict network growth from surficial sediment alone. The prevailing climate sets the frequency and magnitude of precipitation received by a drainage basin, which interacts with the substrate and influences precipitation routing to channels. Coarse-grained, permeable sediments require more frequent or higher magnitude precipitation events to drive overland flow channelization unlike fine-grained, less permeable sediments. If

the climate precludes overland flow, then rivers might develop by seepage erosion if relief is sufficient. A lack of both seepage erosion and overland flow can reduce network growth rates, slowing the development of landscape connectivity. While not accounted for in the experiments, climate also controls the density and type of vegetation within a drainage basin. Vegetation can alter precipitation routing to channels by increasing infiltration and constraining erosion by either process, thus impacting drainage network development.

Climate is particularly important in the Central Lowlands, where climate fluctuations including the transition from glacial to interglacial conditions in the late Pleistocene as well as millennial-scale shifts like the Mid-Holocene Warm Period have affected precipitation. The region's drainage networks generally have wide bifurcation angles associated with groundwater-driven channel development supported by the humid climate (Seybold et al., 2017, 2018). However, the dry tundra environment during the last glaciation provided less precipitation to drive channelization but also had less vegetation to resist erosion and

increase infiltration. Furthermore, permafrost formation in the soil in portions of the Central Lowalnds during glacial periods can inhibit infiltration and drive overland flow (Kasse, 1997). Wetter interglacial periods provide more precipitation to drive channelization, but also increase the amount of vegetation that resists erosion and increases infiltration (Langbein and Schumm, 1958). As the interplay of climate, vegetation, and infiltration change in drainage basins, overland flow or seepage erosion may play a more or less dominant role in channelization through time.

**6. Conclusions**

To gain insight into the processes of channel development in low-gradient landscapes, we conducted small-scale experiments to observe channel development on a low-gradient, internally-drained surface with different rainfall rates, substrate compositions, and a constant rate of base level fall. Several key findings were:

- Many channels underwent a process transition as they evolved [Fig. 5]. Channels that initially formed by overland
flow transitioned to seepage erosion once channel incision generated enough relief to permit flow to channel heads via both surface and subsurface pathways and allow mass wasting via seepage erosion to occur [Fig. 9].

- Landscape variables that mediated infiltration and runoff affected the processes by which channel networks evolved [Fig. 10]. Overland flow was dominant when conditions favored surface water accumulation and routing, namely, when the substrate had a lower infiltration capacity (i.e., more clay) or when the rainfall rate considerably outpaced

infiltration [Table 1; Fig. 10]. Seepage erosion was dominant when the substrate was less cohesive and had a higher infiltration capacity (i.e., less clay): the lower cohesion reduced the force needed for seepage erosion to occur, and higher infiltration capacities allowed more precipitation to enter the subsurface thus increasing the driving force of water exfiltrating to the surface [Table 1; Fig. 10].

- Overall erosional competence of overland flow versus seepage-driven erosion was dependent upon both the areal extent eroding via each process as well as the rate of erosion. For example, overland flow channels eroded greater volumes of sediment [Fig. 14] due to their extensive channelized area [Fig. 9] but had smaller incision rates than seepage erosion [Fig. 13]. Some of the dominance of overland flow erosion can be attributed to conditions early in the runs when overland flow dominated in part due to the initial saturation of the substrate.

- In these experiments, overland flow channels had a larger upland CA compared to seepage erosion, allowing overland flow to integrate more NCA as channels eroded headward [Fig. 11]. Since overland flow was the dominant process throughout most experiments, channels integrated NCA at similar rates under all conditions [Fig. 6, 7].

We considered the implications of these findings for drainage networks in the glaciated Central Lowlands where channels have developed in low-gradient topography by integrating NCA through time. The experimental results suggest that the degree of channel incision, glacial sediment texture, and changing climate likely influence the dominant pathway by which precipitation

reaches channels. In addition, integration of NCA via subsurface flow may draw water from farther away than surface divides suggest, as subsurface divides in low-gradient landscapes may not mimic surface divides. Whether precipitation travels primarily via the surface or subsurface would favor overland flow or seepage erosion at different points in time and space. If past conditions consistently supported overland flow, then channels may have integrated NCA at a relatively constant rate after an early period of channel initiation. However, post-glacial landscapes include vegetation, topographic features, and complex

geology that likely caused the pace of channel development to vary more through time and space than the idealized experiments.

**Data Availability**

All data are available through the University of Minnesota Digital Conservancy (Sockness and Gran, 2021, https://hdl.handle.net/11299/224734) including all raw DEM data as TIFF files and JPEG images as well as analyzed data in

ArcGIS map package files.    Data are also linked to the M.S. thesis of Brian Sockness (2020) at https://hdl.handle.net/11299/213050.

**Author contribution**

BS planned and conducted all experiments and analyses and wrote the manuscript from his M.S. thesis. KG secured funding, advised BS on his M.S. thesis research, and helped with manuscript revisions.

**Competing Interests**

The authors declare they have no conflict of interest.

**Acknowledgements**

The research was funded by a grant from the U.S. National Science Foundation (EAR-1656969). We appreciate the discussions and assistance from Alison Anders, Pete Moore, Bradley Miller, Cecilia Cullen, Nooreen Meghani, and Joshua McDanel and
technical assistance in the laboratory from UMD students Paige Melius, Hannah Schwartz, Elizabeth Boor, and Heidi Krauss. We want to thank Francois Metivier and one anonymous reviewer for improvements to this manuscript.

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
