# Peer review of "An experimental study of drainage network development by surface and subsurface flow in low-gradient landscapes"

_Earth Surface Dynamics, 2021_

## Referee Comment (RC2)

**An experimental study of drainage network development by surface and subsurface flow in low-gradient landscapes by Brian G. Sockness and Karen B. Gran**

F. Métivier

December 14, 2021

**1 General comments**

The paper presents a set of experiments of drainage network development performed on a cylindrical drum equipped with a rainfall simulator and a base level control. The objective is to study the progressive development of channel networks on initially flat "lowland" surfaces, and to discuss qualitatively the respective influence and drivers of sapping and overland flow on channelization and network growth. The paper makes interesting claims yet, as it is, clarifications are needed to sustain them.

1. The paper is biased towards overland flow type processes. The definition of Non Contributing Areas (NCA) does not take into account the fact that groundwater divides can extend far into topographically flat areas. Thus much of the discussion on hydrologic connection applies to surface-water connection only and, although this lowers the ambition of the paper, it should be clearly acknowledged or challenged.

2. Experimental conditions and runs have to be more precisely described especially the way infiltration rates were measured.

3. The authors discuss the difference between two channel types but we lack a description and characterization of these channels. We should be shown various examples of these channel types, their characteristics should be discussed, and so for the evolution from one type of channel to the other.

4. Eventually I would suggest a revised version could be more focused on the experiments, their description and analysis in order to provide a useful account of these runs.

**2 Specific comments**

**17** in the abstract you mention that seepage and overland flow channels occur for different infiltration and precipitation rates, then you tell that seepage occurs after overland flow channels had developed. This is a bit confusing

**20** What do you mean by surface water contributing area for seepage channels ?

**32-35** The notion of NCAs as internally drained areas seems a little bizarre to me. This view seems oriented by some surface flow vision. I can't see how these NCA can't be connected through groundwater flows to the network that is going to incorporate them. Given the setup the entire system has a single groundwater drainage whose boundaries are the cylinder walls.

**55-57** Again this claim only applies to surface flows, At least at the beginning of the experiment.

**60** I would be curious to know how these spillover events have been recorded in natural settings.

**65** Why should hydraulic conductivities be contrasting ?

**Table 1** How do you measure the infiltration capacity $I$ ?

**Table 2** The "low rainfall" rate is $8\,\mu$m/s $\sim 29$ mm/h which turns out to be a **very large** rainfall rate. For an experiment of 10 hours this means a rainfall on order of $30$ cm... This does not often happen in real life even in equatorial settings. The "high rainfall" rate then corresponds to $\sim 60$cm for ten hours which is within a factor of two from the world record for a precipitation of that duration.

**Table 1** It seems as if no erosion and network growth can occur if there is no continuous uplift. Then, how can you decorrelate the importance of this factor compared to the others ? what if there was no uplift or at least if $U/R << 1$ ? Did you test this ?

**Figure 2** How do you prevent water flow from the tank sides and if you do not prevent it how does it affect the experiment ?

**205** What would happen if the system was not saturated before the start ? Given the Infiltration rate you describe I suspect you would have no overland flow and only sapping processes. Did you try ? Did sapping channels form and develop ?

**225** The definition of NCA and CA as zones where *surface* waters do or do not contribute is made clear here. It should probably appear earlier.

**235** Same comment for the influence of flow on the walls. It should appear earlier in the description of the experiment.

**250** Can you show examples of each type of channel head please so we can make up our minds ?

**Table 3** What is the rationale behind these choices and again can you show us the channel types you are talking about ?

**Figure 6** The Channel type 2 you seem to show here looks more like a mass flow then a sapping channel. Do you have other examples ? Are these channel types always localized on the borders ?

**324** You say that type 1 overland flow channels comprise the vast majority of channels during the first half of the experiment but you force this to be so by saturating your media before the start of the experiment. This does not seem to be natural. Then could you show some example pictures of the changes from type 1 to type 2 you describe ? I would really be curious to see how run 3 looks like in the end.

**338** what sort of unit is this $m^2 \cdot m^{-2}$ ? Is that not just the same as a fraction ? Can you explain ?

**Figure 8** There are maybe two issues here that are pervasive in the entire manuscript. First you are not clear about the difference you make between CA aka contributing area and CA aka channelized area (sic). is there any and if so can you be more specific. If, as it seems at first, they are the same then a second problem arises as do not know what is the real basin area of seepage channels. You always stick to your surface flow definition of the contributing area. But this has not much meaning for a channel fed by groundwater. You should be clearer on these issues and potential limitations.

**420 and discussion thereafter** Again because your initial setup favors a specific type of process (overland flow channels) it seems difficult to be able to sort out the influence of the factors you describe with your initial saturation. It really seems important that you discuss this point somewhere.

**455** Again if one looks at table 1 the infiltration rates you give should prevent infiltration because they are much higher then your rainfall rates (which are already huge). Thus this discussion seems a bit far-fetched.

**465-470** I'd love to be persuaded but clearly you do not disclose the evidences needed to claim this.

**499** Again I think that because you saturate your media prior to the start your experiment is biased towards overland flow and therefore conclusions about clay content and rainfall are probably far-fetched.

---

## Author Comment (AC1)

Response to Referee Comments

We would like to formally thank the reviewers for their comments and insight. We have revised our manuscript to take their comments into consideration and feel the manuscript has benefitted greatly from the review process. We address conceptual issues and comments that were in common from both reviewers first, and then address detailed comments from each reviewer.

Both referees sought a more comprehensive review of the experimental design and methodology and results from each experimental run. We have revised the Results and Methodology sections to provide additional details regarding our experimental design (e.g., infiltration capacity measurements and process for simulating uplift), analysis (e.g., method used for channel type classification, calculation of basin integration rates, and calculation of channel incision rates), and results of individual runs versus aggregated observations based on experimental conditions. We also include an additional figure depicting the channel type classification methodology (Fig. 4) and a digital supplement with imagery for all runs like Figure 6 in the current manuscript.

Both referees noted confusion regarding the delineation and definitions of contributing versus non-contributing areas (NCA). Referee #2 highlighted that NCA neglects to consider the fact that groundwater divides can extend far into topographically-flat areas. Our use of the term "non-contributing area" only refers to surface waters, and we clarify that in the first paragraph of the introduction. We completely agree that the groundwater flow to channels can deviate from topographically-defined contributing areas, especially in low-gradient environments, and that areas labeled as NCA at the surface can contribute groundwater to channel heads. In fact, groundwater contributions from areas delineated as non-contributing area were likely critical in driving channel development. The presence of groundwater-driven processes in our experiments underscores the potential deficiencies of viewing NCA in the landscape as hydrologically isolated from observations of the surface topography alone. We have revised the manuscript to try and make it more clear that CA delineated through surface topography only refers to surface contributions and that subsurface divides are not constrained by surface divides.

Referee #2 felt that the manuscript was biased towards overland flow processes for two reasons: 1) contributing areas (CA) and non-contributing areas (NCA) are delineated by surface topography alone, and 2) sediments were saturated prior to the start of the experiments. As noted above, NCA was specifically defined to focus on surface water contributions. One of the main conclusions was that groundwater plays an important role in channel network growth and thus NCA determination as it is traditionally done does not account for this important contribution. We make that more clear now. We do want to be clear that defining CA and NCA based on surface water contributions alone does not bias the experiments towards overland flow processes. The definition of sub-watersheds eroding predominantly via overland flow processes was defined based on slope, relief, and morphology of channel heads, and then compared with upstream CA and NCA. By keeping the definitions of CA and NCA focused on surface water only, it helps us better distinguish the potential drivers of overland flow (i.e. greater CA). Unfortunately, we do have a way to distinguish the subsurface drainage divides and thus cannot compare directly the subsurface contributing area to channel heads that might be connected specifically to seepage erosion.

We also want to address Referee #2's concerns regarding saturating the sediment before initiating our experiments noting that it may bias the results towards more overland flow. This is a really good point, and we acknowledge that it likely did bias the results towards overland flow at the start of each run. Later in the runs, once more relief had been established, the partitioning of flow between overland flow and

subsurface flow as mediated by infiltration capacity and substrate erodibility is more informative, and we still feel that the results later in the runs are not biased by initial conditions.

The decision to saturate the sediment was intentional. Three conditions made it necessary to saturate the sediment before beginning the experiments: (1) infiltration rates exceeded the rainfall rate under all conditions, (2) the initial topography was flat, and (3) no uplift had been applied to the system (i.e., base level drop via lowering the outlet's gate). An unsaturated experiment would likely exhibit an initial lag time with little to no channelization since surface flow processes would be inhibited by the flat topography and high infiltration. Given no initial uplift, seepage erosion would also be precluded because the outlet would lack a seepage face for groundwater to discharge from and entrain sediment. Therefore, to facilitate the immediate formation of channels, we chose to saturate the sediment from the experiment's onset. The referee is correct that initial saturation does bias the experiments toward overland flow at the beginning of the experiments since some amount of initial uplift could have been applied to provide a seepage face. The manuscript has been updated to note this, both in the methods section and in section 5.1 of the discussion.

Several comments concerned scaling relationships between our experiments and the real-world. The model we used was designed to be a process model rather than a scale model, to demonstrate whether varying conditions could lead to different processes of channel development. Previous experiments have focused predominantly on providing ideal conditions for either surface water or groundwater-driven processes of channel development. Our experiments sought to provide a middle-ground with suitable conditions for either process to occur, uniquely demonstrating how both processes could co-evolve in the same low-gradient drainage basin. Although morphologies may be disproportionate to those found in the real-world, we feel the conclusions derived from inferring processes from morphologic indicators, observing the integration of NCA through time, and delineating surface water drainage areas using topography remain valid. Revisions to the manuscript include a new paragraph at the start of the discussion on scaling.

Referee #1 – General Comments (*in italics*)

*"The captions of the figures are not complete. Ideally, a figure should be understandable (at least the main point) just by reading the caption. Often the reader does not [know] which run is tested on each figure."*

> Response: The revised manuscript includes revised figure captions to provide greater context for interpreting and understanding the importance of data presented in each figure.

*"The method to determine the contributing and non-contributing area is not clear for me. A graphical explanation may be useful."*

> Response: The manuscript includes Figure 3 that demonstrates the process of delineating contributing and non-contributing area. The text preceding this figure was revised to better couple with the language and imagery shown in the figure.

*"I do not understand also, what is the quantitative criterion to attribute a channel to overland flow or seepage. The numeration should be used consistently."*

> Response: The current manuscript includes Table 3 listing the quantitative criteria for attributing a channel to overland flow. The criteria were consistently applied across all runs. The revised manuscript includes an additional figure showing how these criteria were applied to classify

channels. In addition, we made it more clear where the cut-off thresholds are for determining if a channel head is mapped as overland flow (Type 1) or seepage (Type 2).

"*Moreover, some of the conclusions are not really supported by the data. Which curve does demonstrate each conclusion point? Some behavior occurs for only one run, so it may be incidental. As the number of runs is small, all should be shown with the same kind of plot found in Fig.6, maybe in appendix*."

Response: The revised manuscript has been updated to relate our conclusions with an associated figure. The authors agree that the chance of incidental behavior is greater due to the small number of runs. Therefore, the revised manuscript now includes an appendix with imagery like Figure 6 to provide readers with the opportunity to independently evaluate each run. In addition, throughout the discussion, we have tried to make it more clear which data support each statement made.

"*The run 1 is closer to the previous experiments of the literature. Do the authors find the amphitheater-headed channels in this run? I would say, that the elevation of the watertable is significantly smaller than the total bed, which produces mass wasting under the form of slumping or sapping events. At least one example of mass wasting should be shown and illustrated*."

Response: The referee is correct that run 1 is the most similar to previous seepage erosion experiments as it was conducted with relatively cohesionless sediment. Yes, amphitheater-shaped channel heads attributed to seepage erosion were observed in run 1, as depicted in Figure 8. Additional imagery included in the digital supplement will demonstrate examples of mass wasting events associated with seepage erosion channels.

"*I do not understand also the discussion of the field examples. In Fig. 13 and 14, in which case seepage erosion is dominant (likely right and left). The morphological indicators must be indicated in the caption. I note, also there is no discussion about the scaling between laboratory experiments and the field. Can we deduce the relevant time and space scales using the laboratory results? Are the shapes similar?*"

Response: The imagery in Figures 13 and 14 has been updated to highlight the key morphological features demonstrated by the field examples. Likewise, the caption will be updated with a brief discussion of the morphological indicators. The scaling issue is addressed up above.

Referee #1 – Specific Comments

1.) "*The uplift process is not sufficiently described and discussed. Can the authors show in the schema of Fig. 2, how this uplift is applied? Consequently, does the main slope evolve with time*."

Response: There are three issues here. The first involves how uplift is accomplished. Uplift of the basin is accomplished by lowering base level. Because that could cause confusion, we have edited the text to make it more clear that we are lowering base level at the outlet. The second issue is how that is being done. This is described in depth on lines 189-194. We have included more detail in the caption for Figure 2. In addition, Figure 2 is revised to show the gate mechanism and associated step-motor used to control base level at the outlet of the system more clearly. The caption and supporting text have been revised to explain how lowering base level at the outlet using the gate mechanism effectively applies uplift to the system. The last issue is whether the main slope evolved with time. The main channel slopes are a function of the amount of base level

fall and the distance of the channel head from the outlet. It is possible is was constant over time, but the main channel slope was not evaluated as part of our analysis.

2.) "*Is Table 3, obtained for a specific run? If not, are the channels head similar for all parameter values? A graphical example would be worthwhile, to understand the procedure.*"

Response: The values presented in Table 3 were extracted from several runs during early, middle, and late timesteps to capture a representative range of values for both channel types. The channels selected for criteria extraction were judged to be the most characteristic examples of seepage erosion or overland flow channels that could then be applied to more ambiguous channels. The revised manuscript includes an additional figure (Fig. 4) demonstrating this procedure.

3) *How the NCA integration rate is defined and then computed from experimental data? Same question for the incision rate?*

Response: The NCA integration rate is defined as the area of NCA converted to CA per hour. The rate was computed by differencing the area classified as CA in the evaluated timestep from the area classified as NCA in the preceding timestep. The resulting area measurement was then divided by the total time between scans to derive a rate. This text has been added into the methods section following the description of how NCA and CA are defined.

The incision rate is defined as the depth of sediment eroded per hour. Differencing the elevation of sequential timesteps provided the depth component of eroded sediment (Figure 11) which occurred over a known length of time between scans, providing a rate. Computing incision rates for overland flow and seepage erosion channels required aggregating an average incision rate for each channel type per timestep. This clarification is now added to the methods section in the paragraph describing volumetric erosion rate calculations.

Referee #2 – General Comments

"*1. The paper is biased towards overland flow type processes. The definition of Non Contributing Areas (NCA) does not take into account the fact that groundwater divides can extend far into topographically flat areas. Thus much of the discussion on hydrologic connection applies to surface-water connection only and, although this lowers the ambition of the paper, it should be clearly acknowledged or challenged.*"

Response: The comment was addressed in more detail up above. We were only able to quantify surface water connections as we only evaluated contributing and non-contributing area as defined by the topography. The authors agree that groundwater divides can extend far beyond surface water divides, especially in topographically flat areas. The implications of this regarding drainage network evolution are discussed in lines 59-69 of the Introduction.

"*2. Experimental conditions and runs have to be more precisely described especially the way infiltration rates were measured.*"

Response: The revised manuscript includes an expanded Methodology section describing how we measured infiltration rates for different sediment compositions.

*"3. The authors discuss the difference between two channel types but we lack a description and characterization of these channels. We should be shown various examples of these channel types, their characteristics should be discussed, and so for the evolution from one type of channel to the other."*

> Response: The revised manuscript includes an expanded discussion of channel type characterization. Specifically, a figure demonstrating the classification process for the two channel types in greater detail is included (Fig. 4). Additionally, a digital supplement with imagery for each run, like Figure 6, has been added to allow readers to independently evaluate each experiment.

*"4. Eventually I would suggest a revised version could be more focused on the experiments, their description and analysis in order to provide a useful account of these runs."*

> Response: The revised manuscript provides a more thorough discussion of the experimental methodology and results of each run.

Referee #2 - Specific Comments

*(Line 17) "in the abstract you mention that seepage and overland flow channels occur for different infiltration and precipitation rates, then you tell that seepage occurs after overland flow channels had developed. This is a bit confusing"*

> Response: The revised Abstract clarifies that seepage erosion and overland flow did not occur exclusively under a given set of conditions. Rather, the experiments had co-evolving channels by both processes, which developed to a greater or lesser extent depending on the experimental conditions and amount of relief generated by channel incision over time.

*(Line 20) "What do you mean by surface water contributing area for seepage channels?"*

> Response: Although groundwater drives seepage erosion, the channels maintain a small surface water contributing area defined by the topography.

*(Line 32-35) "The notion of NCAs as internally drained areas seems a little bizarre to me. This view seems oriented by some surface flow vision. I can't see how these NCA can't be connected through groundwater flows to the network that is going to incorporate them. Given the setup the entire system has a single groundwater drainage whose boundaries are the cylinder walls."*

> Response: As stated in the General Response section, the authors agree that NCAs may not be hydrologically isolated. We specifically define NCAs with respect to surface water contributions, and thus internally-drained means that surface water drains into those areas instead of into the expanding channel network. The water that enters areas defined as NCAs then either infiltrates and flows as subsurface flow into the channel network or flows into the channel network via spillover events as water levels rise. The balance between those two pathways is controlled by both precipitation and infiltration rates. In the natural world, water in NCAs could also be lost through evaporation and transpiration but that is unlikely in our experimental set-up. The impermeable bottom and sides of the tank do direct groundwater to drain at the channels, which increases the likelihood that NCAs are contributing groundwater to channels. These points are clarified in the revised manuscript.

*(Line 55-57) "Again this claim only applies to surface flows, at least at the beginning of the experiment."*

Response: Agreed, the following paragraph, lines 59-69, presents the case for hydrological connections between NCA and channels via surface or groundwater.

*(Line 60) "I would be curious to know how these spillover events have been recorded in natural settings."*

Response: Hilgendorf et al. (2019) summarized the literature regarding spillover events and how they have been recorded in natural settings. Generally, these events are recorded by fluvial systems that flow across topographic barriers (transverse drainages). Spillover is a mechanism by which channels could breach topographic divides and form a persistent outlet. A classic version of spillover events occurs frequently in wetland and lake complexes throughout the glaciated Central Lowlands of the northern U.S. Wetlands or lakes that are internally-drained under most conditions spill into channel networks during high precipitation events that drive levels over low divides. In most cases, those connections are transient, but incision from repeat events can make those connections permanent.

*(Line 65) "Why should hydraulic conductivities be contrasting?"*

Response: The cited studies were conducted in glacial settings with deposits characterized by complex assemblages of heterogeneous sediment. Depositional environments range from gravelly outwash deposits with higher hydraulic conductivities compared to clay-rich glacial tills. Successive glacial advances and retreats deposit layers of sediment with varying compositions, thus contrasting hydraulic conductivities are present at depth.

*(Table 1) "How do you measure the infiltration capacity I?"*

Response: The manuscript has been updated to include the following description of infiltration capacity measurements. We measured infiltration capacity of each sediment composition using a single ring infiltrometer constructed of a 30 cm long cylindrical tube. The tube was placed vertically over a bed of pea gravel to allow for drainage and loaded with sediment to a thickness of 15 cm. After saturating the sediment, water was then added to the tube to a depth (head) of 10 cm. The time needed for the falling head to completely infiltrate the sediment was recorded, allowing an infiltration capacity measurement to be calculated. The test was repeated a dozen times, and the average value was reported in Table 1.

*(Table 2) "The "low rainfall" rate is 8µm/s ~ 29mm/h which turns out to be a very large rainfall rate. For an experiment of 10 hours this means a rainfall on order of 30cm... This does not often happen in real life even in equatorial settings. The "high rainfall" rate then corresponds to ~ 60cm for ten hours which is within a factor of two from the world record for a precipitation of that duration."*

Response: Agreed, even the lowest rainfall rate used in the experiments were greater than rainfall rates found in the most humid environments. However, such rainfall rates are not uncommon when using physical models. Hasbargen and Paola (2000) and Gazzetti (2015) used a nearly identical apparatus with rainfall rates of 6.5 µm/s and 17-24 µm/s, respectively. Other experiments using different apparatuses had similarly high precipitation rates: Lague et al. (2003) (28 µm/s), Ouchi (2011) (10.5 µm/s), and Turowski et al. (2006) (12.5 – 39 µm/s), to name a few. Rainfall rates are high to provide consistent channel-forming discharges that allow drainage networks to form on the order of hours. Although it would be rare to find a storm system that delivered rainfall intensities of 3 cm/hr for 10 hours, it is not uncommon for rainfall intensities of

that magnitude to fall for short durations of time. By keeping rainfall rates high during the entire duration of the experiment, we allow for continuous erosion throughout the experiment.

*(Table 1) "It seems as if no erosion and network growth can occur if there is no continuous uplift. Then, how can you decorrelate the importance of this factor compared to the others? what if there was no uplift or at least if U/R << 1? Did you test this?"*

Response: We did not conduct experiments without uplift or with a lower uplift rate. We expect that a run without uplift would lack channelization given the flat initial topography. The basin would have simply filled with water until spilling over at the outlet with no means of incising channels. Over an equivalent timespan to other experiments, a much lower uplift rate could have resulted in some important differences in channel evolution. Given the importance of channel incision in creating relief throughout the basin and enabling seepage erosion, a lower uplift rate could have generated less relief and inhibited seepage erosion channels from forming. We maintain that since the uplift rate was constant across all runs, we can decorrelate its influence at the given rate.

*(Figure 2) "How do you prevent water flow from the tank sides and if you do not prevent it how does it affect the experiment?"*

Response: Water flow along the tank sides did occur during the experiments. This was mitigated to some extent by placing barriers along the upper lip of the tank to prevent water pooled there from spilling into the basin. However, there was some water flow that incised channels along the edge of the tank during some runs. When this occurred, those channels and associated drainage areas were excluded from analyses. We note this in the revised Methodology section.

*(Line 205) "What would happen if the system was not saturated before the start? Given the Infiltration rate you describe I suspect you would have no overland flow and only sapping processes. Did you try? Did sapping channels form and develop?"*

Response: See our General Response addressing concerns about saturating the media.

*(Line 225) "The definition of NCA and CA as zones where surface waters do or do not contribute is made clear here. It should probably appear earlier."*

Response: Agreed, the revised manuscript notes these definitions earlier.

*(Line 235) "Same comment for the influence of flow on the walls. It should appear earlier in the description of the experiment."*

Response: Agreed, the revised manuscript notes the influence of water flow along the tank walls in the Methodology section.

*(Line 250) "Can you show examples of each type of channel head please so we can make up our minds?"*

Response: The revised manuscript includes a digital supplement with time series data for each run. In addition, all data are in Sockness and Gran (2021), in a public data repository as noted in the Data Accessibility section.

*(Table 3) "What is the rationale behind these choices and again can you show us the channel types you are talking about?"*

Response: The values presented in Table 3 were extracted from several runs during early, middle, and late timesteps to capture a representative range of values for both channel types. The channels selected for criteria extraction were judged to be the most characteristic examples of seepage erosion or overland flow channels that could then be applied to more ambiguous channels. The revised manuscript includes an additional figure demonstrating the channel classification methodology for each channel type. Time series data for all runs are now in a digital supplement.

*(Figure 6) "The Channel type 2 you seem to show here looks more like a mass flow then a sapping channel. Do you have other examples? Are these channel types always localized on the borders?"*

Response: Yes, other runs formed channels via seepage erosion. The revised manuscript now includes an digital supplement with time series data for each run depicting other examples. Seepage erosion channels were not localized to the borders exclusively.

*(Line 324) "You say that type 1 overland flow channels comprise the vast majority of channels during the first half of the experiment but you force this to be so by saturating your media before the start of the experiment. This does not seem to be natural. Then could you show some example pictures of the changes from type 1 to type 2 you describe? I would really be curious to see how run 3 looks like in the end."*

Response: See our General Response addressing concerns about saturating the media. The revised manuscript includes a digital supplement with time series data for each run that depicts the transition from type 1 to type 2 channels.

*(Line 338) "[W]hat sort of unit is this m2 · m−2? Is that not just the same as a fraction? Can you explain?"*

Response: Correct, it is a fraction. We included units to ground fractional values in their physical basis.

*(Figure 8) "There are maybe two issues here that are pervasive in the entire manuscript. First you are not clear about the difference you make between CA aka contributing area and CA aka channelized area (sic). [I]s there any and if so can you be more specific. If, as it seems at first, they are the same then a second problem arises as [we] do not know what is the real basin area of seepage channels. You always stick to your surface flow definition of the contributing area. But this has not much meaning for a channel fed by groundwater. You should be clearer on these issues and potential limitations."*

Response: Contributing and channelized areas are two distinct components of the watersheds in our experiments. Contributing area refers to the upland (non-channelized) area that contributes surface water to channels. Channelized area refers to the area occupied by a channel: It does not include the area upstream of the channel heads that drains into the channelized area. The abbreviation for contributing area, CA, defined in line 224, was used only when referring to contributing area. The revised manuscript: (1) distinguishes between contributing area versus channelized area more clearly and (2) more clearly states that contributing area defined by surface topography does not necessarily represent the groundwater contributing area.

*(Line 420) "and discussion thereafter [referee emphasis] Again because your initial setup favors a specific type of process (overland flow channels) it seems difficult to be able to sort out the influence of the factors you describe with your initial saturation. It really seems important that you discuss this point somewhere."*

Response: See our General Response addressing concerns about saturating the media. The revised manuscript includes a discussion of the implications of saturating the media before initiating the experiment. The main issue is that it may lead to a bias in overland flow at the start of the experiments. Later in the experiments, when there is enough relief for seepage erosion to start occurring, the initial saturation of the basin likely did not have an impact on erosional rates. We have clarified that in the discussion now.

*(Line 455) "Again if one looks at table 1 the infiltration rates you give should prevent infiltration because they are much higher then your rainfall rates (which are already huge). Thus this discussion seems a bit far-fetched."*

Response: We assume that "prevent infiltration" is an error, and the referee intended to state "prevent overland flow/runoff/surface flow." We disagree that infiltration rates exceeding rainfall rates preclude overland flow, which is described by the process of saturation overland flow (i.e., saturation from below). Given saturated sediment, the water table is near the surface throughout the basin and is nearly coincident with the channel surface. Constant rainfall across the basin raises the water table as it infiltrates, causing the water table to reach the surface, especially in near-channel areas. Additional water input to these saturated areas travels as overland flow. Increasing or decreasing the clay content scales the degree to which this occurs in addition to modifying sediment cohesion.

*(Line 465-470) "I'd love to be persuaded but clearly you do not disclose the evidences needed to claim this."*

Response: The conclusions regarding rainfall rates, clay content, and erosion volumes are supported by empirical measurements. The conclusions regarding groundwater flow patterns are more interpretive because we had no means of measuring groundwater flow in our experiments. However, the presence of active seepage erosion coupled with small surface contributing areas support the role of groundwater flow in driving channel network development even though we could not measure the subsurface contributing areas directly.

*(Line 499) "Again I think that because you saturate your media prior to the start your experiment is biased towards overland flow and therefore conclusions about clay content and rainfall are probably far-fetched."*

Response: See our General Response addressing concerns about saturating the media.

---

## Author Response (AR2)

Response to Reviewers:

We thank the reviewers for their continued efforts to improve the manuscript. As Reviewer 2 found our revised manuscript acceptable for publication, here we primarily address on-going concerns from Reviewer 1.

The most important concern raised is that the areas with erosion due to seepage erosion vs. overland flow are delineated morphologically. Without additional data proving that Type 1 morphology was caused only by overland flow and Type 2 morphology was formed only due to seepage erosion, we are not able to make that case. This is an important concern, and we address it below as best we can.

Below are the Reviewer's comments related to this followed by our response:

R1: *I have a major concern about this work. The attribution of the contributing area to overland or seepage flows relies only on the morphological distinction between channels of type 1 of low slope and channels of type 2 of high slope. The firsts are attributed to overland or runoff flows and the seconds to seepage erosion through sapping events. Here with the presented data we cannot determine if this assumption is correct. Moreover, I note that for seepage erosion experiments with an inclined granular bed, the upper channels display also a branching shape of low slope with the merging of several sub-channels into a larger stream. See "Lobkovsky, A. E., Jensen, B., Kudrolli, A., & Rothman, D. H. (2004). Threshold phenomena in erosion driven by subsurface flow. Journal of Geophysical Research: Earth Surface, 109(F4)", "Smith, B., Kudrolli, A., Lobkovsky, A. E., & Rothman, D. H. (2008). Channel erosion due to subsurface flow. Chaos: An Interdisciplinary Journal of Nonlinear Science, 18(4), 041105", and "Schorghofer, N., Jensen, B., Kudrolli, A., & Rothman, D. H. (2004). Spontaneous channelization in permeable ground: Theory, experiment, and observation. Journal of Fluid Mechanics, 503, 357-374".*
*Please comment.*

*The authors must prove that the channels of type 1 are really created by the erosion of overland flows only. Otherwise the conclusions of the article are not supported by the data.*
*If not; the bed inclination would be caused here by the imposed uplift. High slope channel heads created by sapping would occur when the height of the granular bed above the emergence of the water stream exceeds a certain value and would correspond to a particular case of seepage erosion at the end of experimental runs. As the infiltration rate is always smaller than the precipitation rate, I suppose there is always infiltration and a non-negligible contribution of seepage erosion, even when the bed is initially saturated in water. Then, some of the non-contributing area can participate to the filling of the water table and thus to the seepage erosion. Please comment.*

> We recognize the reviewer's concerns that our delineations of Type 1 and Type 2 channel heads are based solely on morphology, with the interpretation that Type 1 are derived from overland flow and Type 2 are derived from seepage erosion. Reviewer 2 was concerned that our experiments were biased towards overland flow because the experiment began fully saturated. Reviewer 1 has the opposite concern, that erosion may

be dominated by seepage erosion throughout, and that we need to demonstrate conclusively that overland flow alone was responsible for Type 1 erosion. They reference a series of experiments investigating seepage erosion in granular material (Lobkovsky et al., 2004; Smith et al., 2008; Schorghofer et al., 2004).

Reviewer 1 states that we "*must prove that the channels of type 1 are really created by the erosion of overland flows only*". We cannot do this, because we do not think the channels were eroded by overland flow alone. Erosion is unlikely to be unimodal, driven only by overland flow or by seepage erosion. The conditions in our experiments allowed for both processes to occur.

The Reviewer expresses concerns that our "*attribution of the contributing area to overland or seepage flows relies only on the morphological distinction between channels of type 1 of low slope and channels of type 2 of high slope.*" We want to note that Type 1 and Type 2 channel heads have quite distinct morphologies. Measurements were made of prototypical examples of each type of channel head, and those data were then used to set the threshold criteria for classifying all channel heads as either Type 1 or Type 2 based on both local slope and relief at the channel head. Those criteria are listed in Table 3. It is misleading to state that our attribution relies only on low slope vs. high slope.

To help illustrate the differences between the two channel head types, we included an additional figure (Fig. 4) showing a photo of one of the experiments with both Type 1 and Type 2 channel heads. In this revision, we have also added in a figure in the Supplemental file showing channel profiles for both Type 1 and Type 2 channel heads from the last time step in all six runs. Channel profiles are generally linear except near the channel head where they are concave and steep on Type 2 profiles and linear all the way to the channel head on Type 1 profiles. They are quite distinct.

In order to map them in a systematic way, we employed thresholds of both local slope and relief, using data collected from prototypical examples of the two channel types. In places where overlaps occurred, they are mapped as Type 1, thus there is a bias towards Type 1 channel heads because of this. We openly acknowledge that in the paper.

We have tried to be very clear in our paper that we are interpreting the different morphologies as erosion due primarily due to overland flow and seepage erosion. It is stated as an interpretation. Since we did not directly measure flow pathways during the experiments, it will have to remain as an interpretation. The interpretation is based on our observations, experimental conditions, and observations from other studies in the literature.

First our observations. The experiments officially began once the system was fully saturated and overland flow was occurring. This is when base level fall was initiated, providing some relief in the system. Although the infiltration capacity exceeded the precipitation rate, infiltration capacity is different than infiltration rate. When the subsurface saturated, no more water could infiltrate. Thus, we had observable saturation overland flow in all runs. Runs with a higher infiltration capacity could move water

through the subsurface faster and thus infiltrate greater amounts of rainfall.  Runs with higher precipitation rates or lower infiltration capacities would have a greater fraction of the precipitation flowing across the surface.  The balance between overland flow and subsurface flow thus depended on the ratio between infiltration capacity and rainfall rate.  Both surface and subsurface flows occurred in all runs, and erosion via both overland flow and sapping occurred in all runs.  We are interpreting the dominance of one vs. the other at the channel heads based on their morphology.

Second, comparisons with the literature.  The papers by Lobkovsky et al. (2004), Schorghofer et al. (2004), and Smith et al. (2008) all have similar experimental set-ups, with a sloping surface and a measurable groundwater table producing sapping either in the form of channelized flow or mass wasting.  The channels can have a channelized form, often with a rounded channel head, which is quite different in appearance from the channel heads we labeled Type 1.  The amphitheater-shaped channel heads we labeled as Type 2 closely resemble the mass wasting channel heads found in the experiments with higher water tables.  This is consistent with the development of more Type 2 channel heads as relief developed in the experiments.  We feel confident that the Type 2 channel heads were predominantly eroded from seepage erosion. We do not see our Type 1 channels in the seepage erosion channel examples given in the cited studies by Lobkovsky et al. (2004), Schorghofer et al. (2004), and Smith et al. (2008).  This may be because those experiments had no surface flow.

An additional study relevant here is Berhanu et al. (2012), which is similar to the other ones referenced above.  Berhanu et al. starts with a flat surface rather than a sloped one and includes both rainfall and introduction of water from the subsurface. Their parameters were established to allow for erosion via seepage erosion only, not overland flow, and the channel morphologies they developed are quite different from the channel heads we labeled as Type 1.  They found noticeable and measurable differences in channel head morphology based on how water was introduced to the system (from below only vs. combined with rainfall).  The channel heads developed via seepage erosion in the presence of rain are even less like our Type 1 channels than some of the examples in the experiments of Lobkovsky et al. (2004), Schorghofer et al. (2004) and Smith et al. (2008).

We feel confident that our channels began through erosion via overland flow based on initial experimental conditions and observations of overland flow.  We feel confident that our Type 2 channel heads were eroding via seepage erosion based on morphology of the channel heads, comparisons with literature, and experimental conditions that allowed for substantial subsurface flow to occur.  We have added in additional citations in our manuscript to the papers referenced in this discussion as they help support the interpretation of Type 2 channel heads as developing from seepage erosion.  Although branching channels were found in seepage erosion channels of Lobkovsky et al. (2004), they have quite different morphologies compared to the channel heads we labeled Type 1.  Given the initial conditions of full saturation and our observations of overland flow, we

stand by our interpretation that these channel heads were eroded primarily via overland flow.

Berhanu, M., Petroff, A., Devauchelle, O., Kudrolli, A. and Rothman, D.H., 2012. Shape and dynamics of seepage erosion in a horizontal granular bed. *Physical Review E*, *86*(4), p.041304.

Other comments:

*In the previous experiment and field studies about seepage erosion, the average slope (here due to the uplift) is a key parameter, which is not here sufficiently discussed to my opinion. For example the experimental channels have very different shapes between Lobkovsky 2004 and Berhanu 2012 (case without rain).*

We have gone back to the experimental data and extracted longitudinal profiles from the last time step for each run. Those are now included in Supplemental Figure S7. Channel slopes are dependent in part on the depth of incision and distance from outlet of the channel head. Channel profiles are generally linear except near the channel head where they are concave and steep on Type 2 profiles and linear all the way to the channel head on Type 1 profiles. Slopes on the channels downstream from the channel heads range from 0.11 to 0.22, steep enough for both processes of overland flow and seepage erosion to occur.

*1) Abstract: The sentence: "Seepage-driven erosion was favored in substrates with higher infiltration rates, while overland flow was more dominant in experiments with high precipitation rates, although both processes occurred in all runs. » is not clear and even contradicts itself.*

We disagree that this statement is contradictory. Substrates with high infiltration rates had more Type 2 erosion than substrates with low infiltration rates. Experiments with high precipitation rates had more Type 1 erosion than experiments with low precipitation rates. All runs had channels with both Type 1 and Type 2 channel heads at some point during the run. Thus, both processes occurred in all runs. We do not think that overland flow and seepage erosion are mutually exclusive. Surface water can be contributing to erosion via overland flow, while subsurface flows may be contributing to erosion via seepage erosion. Both processes could thus be occurring simultaneously. Our interpretation is that when one type dominates over the other, it can lead to different morphologies in the channel heads.

*2) The data processing of the DEM is not sufficiently specified. Several tools, like "sinks", "watersheds", "Basin" are evoked but not defined. Are they specific to the commercial software FARO® SCENE? It would be better to define the mathematical operations and precise then the "tool" of the software or the algorithm. I suppose that the software finds the local minima in the elevation data after removing the average slope. Am I right ?*

These terms are specific to ArcGIS software. We have gone back through the manuscript to use more general descriptions of the algorithms, followed by the name of the tools used in the ArcGIS software and made it clear that these are ArcGIS algorithms.

*3) The interpretations at the grain scale are missing. How the cohesion due to the clay modifies the erosion threshold (Shields criterion for example)? Here the addition of clay is only seen as a way to decrease the infiltration rate and favor overland flow. The increase of the erosion volume with the clay at the end of experiments is not really explained.*

We did state that the clay plays two roles: it decreases infiltration and increases cohesion which impacts the erodibility of the material. Although we have measurements of infiltration capacity, we did not make direct measurements of how clay modifies cohesion and critical shear stress. However, other experiments that use sand-kaolinite mixtures do have information on how much cohesion or yield strength was increased with additions of kaolinite clay. We summarize this briefly in the paper and provide more information below.

Reddi and Bonala (1997) investigated the impact of kaolinite clay additions to fine sand of between 10% and 40% kaolinite. They subjected the mixtures of clay, sand, and water to a variety of tests. We took their cohesion measurement data for mixtures, fit a curve to the 10%, 20% and 40% kaolinite data and extrapolated it back to 0%, 2%, and 6% kaolinite. For all mixtures, we fixed 0% clay to be 0% cohesion. For "wet of optimum" conditions, the cohesion for 2% clay was between 1.4 and 3.5 kPa; for 6% clay, the cohesion was between 6.0 and 10.4 kPa. We thus predict that the cohesion in the 6% clay run would be 3 to 4 times as high as in the 2% run. The range is based on the type of regression used. Reddi and Bonala (1997) found a linear relationship between cohesion and critical shear stress, thus changes in cohesion are likely reflected in changes in critical shear stress.

Both Marr et al. (2001) and Ilstad et al. (2004) used sand-kaolinite-water mixtures to model subaqueous debris flows. Ilstad et al. (2004) measured the yield strength of mixtures between 5% and 32.5% kaolinite, with an exponential relationship between the clay content and the stress required for deformation to occur. Extrapolating that relationship to our clay contents (0%, 2%, and 6%) would give a yield stress of 3.4 Pa for the 0% clay content, 4.3 Pa for 2% and 7.2 Pa for 6%. Yield stress should be sensitive to both the sand grain size and the water content. Ilstad et al. (2004) used 500 micron sand, while Marr et al. (2001) used 110 micron sand, similar to our experiments. The water contents in Ilstad et al. (2004) of 35% were similar to our mixtures at the start of the experiments when fully saturated. Because we do not have a perfect match between sand grain size, water content, and kaolinite content, we cannot directly use estimated yield strengths from these publications, but the difference in yield strength may be similar. Thus, based on Marr et al. (2001) and Ilstad et al. (2004) data, we expect that the 2% clay content runs would have a yield strength ~30% higher than 0% clay runs. The 6% clay runs would have a yield strength ~120% higher than 0% clay runs. The yield strength is directly related to the critical shear stress required to erode sediment.

Ilstad, T., Elverhøi, A., Issler, D., & Marr, J. G. (2004). Subaqueous debris flow behaviour and its dependence on the sand/clay ratio: a laboratory study using particle tracking. *Marine Geology*, *213*(1-4), 415-438.

Marr, J. G., Harff, P. A., Shanmugam, G., & Parker, G. (2001). Experiments on subaqueous sandy gravity flows: the role of clay and water content in flow dynamics and depositional structures. *Geological Society of America Bulletin*, *113*(11), 1377-1386.

Reddi, L. N., & Bonala, M. V. (1997). Critical shear stress and its relationship with cohesion for sand. kaolinite mixtures. *Canadian geotechnical journal*, *34*(1), 26-33.

*4) Figure 14. Please increase the size of the symbols.*

We made the symbols smaller on that figure on purpose, so that the standard deviations could be visible. The larger symbols hid the error bars. At your request we went back to the larger symbols and noted in the caption that the error bars are smaller than the symbols.

*5) The implication in the field is interesting in part 5, but remains largely speculative due to the lack of quantitative comparisons to my opinion. However, in the conclusion theses implications are satisfyingly presented as "possible".*

The comparisons with the field are intended to spark reflection on how multiple drivers may be contributing to erosion at the channel heads depending on substrate and relief. As you note, we treat this as "possible" rather than as proven.

---

## Author Response (AR3)

The requested edit to remove the reference to Berhanu et al. (2012) from line 522 has been made.